# Distributionally Robust Bayesian Optimization: From Single to Multiple Objectives

## Abstract

In many real-world applications, systems are typically expensive to evaluate and influenced by contextual variables whose distributions may shift between training and deployment. While robust Bayesian optimization methods have been proposed for black-box functions under such conditions, most of them focus solely on single-objective settings. In practice, however, systems often need to be optimized across multiple criteria simultaneously, which is challenging since the same environment may affect different objectives in distinct ways. Although robustness against the contextual uncertainty has been investigated for single-objective problems, its extension to multi-objective optimization (MOO) problems remains limited, with existing works primarily addressing only input noise—a special case of the contextual uncertainty. To bridge this gap, in this work, we propose the first Multi-objective Bayesian Optimization (MOBO) method for the general $\varphi$-divergence Distributionally Robust Optimization (DRO) problem with shared contexts, aiming to obtain *robust efficient* solutions. Furthermore, a provable regret bound is provided, which is the first sublinear regret bound without requiring a decreasing radius of the DRO uncertainty set, even in comparison to existing works in the single-objective setting. Moreover, we provide numerical experiments to validate our theory and the empirical effectiveness of our proposed algorithms.

## 1 Introduction

Bayesian Optimization (BO) (Kushner, 1964; Jones et al., 1998; Shahriari et al., 2015; Srinivas et al., 2009) is a popular framework to find optimal solutions for black-box functions, which are expensive to evaluate. BO finds a lot of practical applications, such as the hyperparameter tuning for machine learning algorithms (Snoek et al., 2012), chemical space exploration (Hernández-Lobato et al., 2017), and robot planning and exploration under uncertainty (Martinez-Cantin et al., 2007).

However, in many real-world applications, the goal is to optimize multiple, possibly competing, black-box functions rather than a single one. For example, vehicle design requires balancing weight, acceleration, and toe-board intrusion (Liao et al., 2008); material selection involves trade-offs among cost, mass, volume, power-to-weight ratio, and energy density (Ashby, 2000); and drug discovery considers factors such as solubility, toxicity, and potency (Paria et al., 2020). Because these objectives inherently conflict, improving all of them simultaneously is generally infeasible. The Multi-Objective Optimization (MOO) framework addresses this challenge by identifying the Pareto Frontier (PF)—a set of optimal trade-offs among objectives and their corresponding solutions.

In many practical applications, black-box functions depend not only on system parameters but also on context parameters (Krause & Ong, 2011; Feng et al., 2020). For instance, Tesch et al. (2011) study policy learning in a robotics setting where the context changes over time. A key challenge in such scenarios is that the context distribution observed during the training may differ from the true distribution encountered at deployment, leading to significant performance degradation. To address this challenge, the DRO framework (Ben-Tal et al., 2013; Shapiro, 2017; Rahimian & Mehrotra, 2022; Zhang et al., 2025; 2024) is widely adopted. DRO constructs an uncertainty set of distributions centered around the training distribution and seeks solutions that remain robust under the worst-case distribution within this set. However, most existing DRO studies focus on the single-objective setting and rely on gradient-based methods. A few recent works (Kirschner et al., 2020; Husain et al., 2023; Huang et al., 2024; Tay et al., 2022) explore Bayesian Optimization approaches to robustify DRO,

but these too are limited to single objectives. On the multi-objective side, Daulton et al. (2022) investigate robust MOBO, but their formulation only accounts for input noise and measures the robustness via Value-at-Risk (VaR). The only work addressing the MOBO problem under a DRO setting is the initial version of Inatsu et al. (2024), which assumes mutual independence among the contexts of different objectives. While simplifying the analysis, this assumption risks producing overly permissive solutions.

In this paper, we study the MOBO problem, considering a more general robustness towards the distribution shifts of context variables. Our specific contributions are summarized as follows:

- It is the first work to study the robust MOBO problem with the robustness towards the uncertainty in the shared context, rather than input noise. Aiming to obtain robust efficient solutions (Ehrgott et al., 2014), we propose a novel algorithm using random scalarization (shown in Algorithm 1). Our method integrates a wide range of $\varphi$-divergence DRO uncertainty sets, including the well-known TV, $\chi^2$, KL, and Cressie-Read family of divergences.

- For our proposed method, we establish a provable sublinear upper bound on the regret, which quantifies the performance gaps between our obtained solutions and the robust efficient solutions. Our results also extend directly to the single-objective setting. Even compared with existing studies on single-objective problems, our analysis holds under weaker and more practical assumptions: (1) We eliminate the bounded function values requirement imposed by Kirschner et al. (2020); Tay et al. (2022). (2) With a mild technical cost, the additional condition that the reference distribution assigns positive probability to all context values, we relax the condition of a shrinking uncertainty radius (Huang et al., 2024; Tay et al., 2022) by allowing it to remain constant, thereby ensuring that our obtained robust solutions remain effective under distributional shifts in context variables.

- We validate our algorithm through numerical experiments, which include synthetic functions and one real-world problem in single/multi-objective settings.

## 1.1 RELATED WORK

A wide range of robust optimization methods have been studied. Some works focus on the robustness under input noise, e.g., VaR (Duffie & Pan, 1997; Wipplinger, 2007; Pritsker, 1997) and Conditional Value at Risk (CVaR) (Rockafellar et al., 2000; Shapiro et al., 2021; Rockafellar & Uryasev, 2002), while others address the distributional shifts in the context variable (Levy et al., 2020; Shapiro, 2017). The latter setting is more general and includes the first one as a special case. For black-box functions that are expensive to evaluate, many studies have integrated BO methods (Bogunovic et al., 2018; Picheny et al., 2022; Cakmak et al., 2020; Nguyen et al., 2021a;b) to explore the robustness against input noise. However, these methods do not consider the distributional uncertainty in the context parameters. Recently, Husain et al. (2023); Kirschner et al. (2020); Huang et al. (2024); Tay et al. (2022) study the DRBO problem. For example, Kirschner et al. (2020) consider a setting where the objective function lies within a known Reproducing Kernel Hilbert Space (RKHS). An uncertainty set of distributions is constructed via the Maximum Mean Discrepancy (MMD). Nevertheless, the algorithm requires solving a complicated minimax problem, which requires extra computations. Husain et al. (2023) study the DRBO problem with $\varphi$-divergence, while Tay et al. (2022) explore various DRO uncertainty sets. To reduce the computations used for solving the DRO problem, an efficient method is proposed (Tay et al., 2022) to approximate the performance under the worst-case distribution. However, this method introduces an estimation error into the regret bound. Moreover, both Husain et al. (2023); Tay et al. (2022) require the function value to be bounded, which may not necessarily hold in the Bayesian setting. Huang et al. (2024) study the DRBO problem with a continuous context distribution. A kernel-based method is proposed to estimate the underlying context distribution, and the regret for TV-DRO is theoretically characterized. However, for all methods mentioned above, to achieve a sublinear regret, the radius of the uncertainty sets is required to decrease with time. Though this assumption may be reasonable in the data-driven setting, where the estimated distribution becomes more accurate with more data, it is of more interest to study the case where the radius of the uncertainty sets is a constant. The solutions obtained under a constant uncertainty radius remain robust even when the test context distribution differs from the training distribution. Moreover, all these methods (Husain et al., 2023; Kirschner et al., 2020; Tay et al., 2022; Huang et al., 2024) are limited to the single-objective setting. In contrast, this paper considers the multi-objective DRO problem with a constant uncertainty set radius.

| Method | Robustness | Objective | Radius [1] | Sublinear Regret [2] |
|---|---|---|---|---|
| Kirschner et al. (2020) | MMD-DRO | Single | $\mathcal{O}(\frac{1}{\sqrt{t}})$ | ✓ |
| Tay et al. (2022) | MMD, TV, $\chi^2$, Wasserstein-DRO | Single | $\mathcal{O}(\frac{1}{\sqrt{t}})$ | ✗ |
| Huang et al. (2024) | TV-DRO | Single | $\mathcal{O}(\frac{1}{\sqrt{t}})$ | ✓ |
| Daulton et al. (2022) | MVaR | Multiple | N/A | N/A |
| Inatsu et al. (2024) [3] | general risk/DRO | Multiple | N/A | ✓ |
| **This paper** | **general $\varphi$-divergence DRO** [4] | **Multiple** | **Constant** | ✓ |

Table 1: Comparison with settings and results in existing analyses. Explanation of the upper footmarks: 1 : We specify the order of the corresponding uncertainty set radius at time $t$ to obtain a regret in the order of $\mathcal{O}(\sqrt{t})$. 2 : ✓ indicates that the model is able to obtain the sublinear regret with the corresponding radius requirements, and ✗ indicates the model fails to obtain the sublinear regret even when the radius of the uncertainty set is $0$. 3: Inatsu et al. (2024) assume that the context for each objective is independent, allowing the identification of the Pareto frontier of the worst-case vector and providing a distance-based convergence guarantee. 4: Our method works for the $\varphi$-divergence DRO (see equation 6) when $M = \sup\{m : \varphi(m) \leq a, m \geq 0\}$ exists, which includes a lot of DRO problems such as the TV, CVaR, Cressie-Read family with $k > 1$, $\chi^2$-divergence and KL-divergence DRO problems.

Multi-objective optimization (MOO) has been extensively studied and finds applications across diverse domains (Xiao et al., 2024; Désidéri, 2012; Caruana, 1997; Ruder, 2017; Zhang & Yang, 2018). For black-box settings with multiple objectives, multi-objective Bayesian optimization (MOBO) methods have been proposed (Suzuki et al., 2020; Paria et al., 2020; Chowdhury & Gopalan, 2021; Ozaki et al., 2024), though most approaches overlook robustness considerations. Existing robust MOBO studies remain limited. For instance, Daulton et al. (2022) examine robustness under input noise by leveraging the Multivariate Value-at-Risk (MVaR) criterion (Prékopa, 2012), and optimize the hypervolume of the resulting MVaR set. To reduce computational burden, Chebyshev scalarization (Miettinen, 1999) is used to map the vector-valued problem into a scalar one. While their formulation focuses on input perturbations, Inatsu et al. (2024) investigate robustness with respect to contextual uncertainty. Their method assumes that the contexts of different objectives are independent, and optimizes performance under the worst-case context for each objective separately. However, this assumption may yield overly permissive solutions. In contrast, our work focuses on robustness to context variables in a more realistic setting where all objectives share the same input and context environment. This setting naturally subsumes the case of noisy inputs, but poses greater challenges since robustness must be ensured jointly across objectives. Table 1 summarizes the key assumptions and contributions of related studies in comparison with ours.

## 2 PRELIMINARIES

### 2.1 GAUSSIAN PROCESS

**Single-objective setting.** We begin by introducing a single-objective black-box reward function $f : \mathcal{X} \times \mathcal{C} \to \mathbb{R}$, where $x \in \mathcal{X} \subseteq \mathbb{R}^d$ is the model parameter and $c \in \mathcal{C} \subseteq \mathbb{R}^{d'}$ is the context variable. At time $t$, the context variable is $c_t$, and for a given model parameter $x_t$, the current observation is $y_t = f(x_t, c_t) + \eta_t$, where $\eta_t \sim \mathcal{N}(0, \sigma_f^2)$ is the noise and $\sigma_f^2$ is the constant noise variance. The following GP (Williams & Rasmussen, 2006; Krause & Ong, 2011; van Bueren et al., 2021) assumption is widely used in the BO literature: $f \sim GP(m, k)$, where $m : \mathcal{X} \times \mathcal{C} \to \mathbb{R}$ is a known mean function and $k : \mathcal{X} \times \mathcal{C} \times \mathcal{X} \times \mathcal{C} \to \mathbb{R}$ is a known covariance function. We follow the widely used assumptions that $m \equiv 0$ and $k((x, c), (x, c)) \leq 1$ for any $x \in \mathcal{X}$ and $c \in \mathcal{C}$.

**Multi-objective setting.** The objective function is a vector $\boldsymbol{f}(x, c) = [f_1(x, c), ..., f_K(x, c)]$ instead of a scalar where $f_i : \mathcal{X} \times \mathcal{C} \to \mathbb{R}$ is the objective for $i \in [K]$ and $K$ is the number of objectives. We assume that for each $i$ under context $c_t$ at time $t$, the observation is under Gaussian noise as $y_t^i = f_i(x_t, c_t) + \eta_t^i$, where $\eta_t^i \sim \mathcal{N}(0, \sigma_f^2)$ and $\sigma_f^2$ is the constant variance of the observation noise. For each $i \in [K]$, the objective follows a GP: $f_i \sim GP(m_i, k_i)$, where $m_i : \mathcal{X} \times \mathcal{C} \to \mathbb{R}$ is a known mean function and $k_i : \mathcal{X} \times \mathcal{C} \times \mathcal{X} \times \mathcal{C} \to \mathbb{R}$ is a known covariance function. Similar to the single-objective setting, we assume that $m_i \equiv 0$ and $k_i((x, c), (x, c)) \leq 1$ for any $x \in \mathcal{X}$ and $c \in \mathcal{C}$. For

each $i \in [K]$, defined by $\boldsymbol{y}_t^i = [y_1^i, ..., y_t^i]^\top$ the observation vector and $A_t = \{(x_1, c_1), ..., (x_t, c_t)\}$ the dataset at time $t$. Given $A_t$ and $\boldsymbol{y}_t^i$, $f_i(x, c)$ follows a GP distribution. Then the corresponding mean $\mu_t^i(x, c)$ and the variance $(\sigma_t^i(x, c))^2$ can be expressed as:

$$
\begin{aligned}
\mu_t^i(x, c) =& \boldsymbol{k_t^i}(x, c)^\top (\boldsymbol{K_t^i} + \sigma_f^2 \boldsymbol{I})^{-1} \boldsymbol{y}_t^i, \\
(\sigma_t^i(x, c))^2 =& k_i([x, c], [x, c]) - \boldsymbol{k_t^i}(x, c)^\top (\boldsymbol{K_t^i} + \sigma_f^2 \boldsymbol{I})^{-1} \boldsymbol{k_t^i}(x, c),
\end{aligned}
\tag{1}
$$

where the kernel vector is defined as $\boldsymbol{k_t^i}([x, c]) = [k_i([x_1, c_1], [x, c]), \ldots, k_i([x_t, c_t], [x, c])]^\top$ and let $\boldsymbol{K_t^i}$ be the positive definite kernel matrix $[k_i([x_a, c_a], [x_b, c_b])]_{a,b \leq t}$.

## 2.2 DISTRIBUTIONALLY ROBUST OPTIMIZATION

The DRO framework has been well studied for single-objective problems (Ben-Tal et al., 2013; Shapiro, 2017). The motivation is based on the observation that the test distribution is not necessarily identical to the training distribution. Thus, only optimizing the expected loss under the training distribution may lead to significant performance degradation. The key idea of DRO is to construct an uncertainty set of distributions centered at the training distribution and to optimize the expected loss under the worst-case distribution. The uncertainty set of distributions is defined as

$$
\mathcal{U}(P_t) := \{Q | D(Q, P_t) \leq \varepsilon_t\},
\tag{2}
$$

where $P_t$ is the reference distribution, $D$ is a distance measure and $\varepsilon_t \in \mathbb{R}$ is the radius. For the single-objective setting, DRO (Huang et al., 2024; Tay et al., 2022) aims to optimize the objective:

$$
\sup_{x \in \mathcal{X}} \inf_{Q \in \mathcal{U}(P_t)} \mathbb{E}_{c \sim Q}[f(x, c)],
\tag{3}
$$

In this paper, we consider a general setting with a continuous action set $\mathcal{X}$ and a discrete context set $\mathcal{C}$ of bounded cardinality $|\mathcal{C}|$.

However, it is nontrivial to extend equation 3 to the multi-objective case, where $\boldsymbol{f}$ is vector-valued rather than scalar. In the empirical risk minimization setting, the optimization objective is given by $\sup_{x \in \mathcal{X}} \mathbb{E}_{c \sim P_t}[\boldsymbol{f}(x, c)]$, which is well defined. We say that $x_1$ *dominates* $x_2$ if for every $i \in [K]$ we have $\mathbb{E}_{c \sim P_t}[f_i(x_1, c)] \geq \mathbb{E}_{c \sim P_t}[f_i(x_2, c)]$, and the inequality is strict for at least one $i$. The goal of MOO is to identify a *Pareto optimal* point $x \in \mathcal{X}$ such that no other feasible point dominates $x$. In contrast, in the DRO setting, the max–min operation is not directly applicable to vector-valued objectives. To address this, we adopt the notion of the *robust efficient point* (Ehrgott et al., 2014). For each $x \in \mathcal{X}$, define the feasible set by

$$
\boldsymbol{f}_{\mathcal{U}}(x) = \{\mathbb{E}_{c \sim Q}[\boldsymbol{f}(x, c)] : Q \in \mathcal{U}\}
\tag{4}
$$

and $\mathbb{R}_{\geq 0}^K = \{x \in \mathbb{R}^K : x \neq 0 \text{ and } x_i \geq 0, \forall i \in [K]\}$. For sets $A, B$, the operator "+" used below denotes the Minkowski sum, defined as $A + B := \{a + b | a \in A, b \in B\}$. We then have:

**Definition 1.** $\bar{x} \in \mathcal{X}$ *is called robust efficient if there exists no other $x \in \mathcal{X}$ such that*

$$
\boldsymbol{f}_{\mathcal{U}}(x) \subseteq \boldsymbol{f}_{\mathcal{U}}(\bar{x}) + \mathbb{R}_{\geq 0}^K.
\tag{5}
$$

Definition 1 indicates that a solution $\bar{x}$ is assumed to be worse than the solution $x$ if and only if for every distribution $Q \in \mathcal{U}$, there exists a distribution $\bar{Q} \in \mathcal{U}$ under which $\bar{x}$ attains a worse objective value than $x$ under $Q$. Thus, the goal of DRO in the multi-objective setting is to identify a set of robust efficient solutions.

**Comparison with data-driven setting:** Existing works on single-objective DRBO problems (Kirschner et al., 2020; Huang et al., 2024; Tay et al., 2022) consider the data-driven setting, which assumes that there exists a fixed but unknown underlying distribution $P^*$, where the context variable $c \sim P^*$. At each time $t$, based on the historical observation, an estimated distribution $P_t$, e.g., the empirical distribution, is constructed. An uncertainty set of distributions centered at $P_t$ is built, and the objective is optimized under the worst-case distribution in this uncertainty set. Due to the fact that the distance between the underlying distribution $P^*$ and its estimate $P_t$ decreases with time $t$, the radius of the uncertainty set is designed to shrink over time. However, existing studies offer no detailed guidelines for choosing this radius. Moreover, as $t$ increases, the estimated distribution converges to the true distribution, causing the uncertainty radius to shrink toward zero. Consequently,

the solutions obtained from data-driven DRO approaches approach those optimized directly under $P^*$, which lack robustness when the test-time distribution deviates from $P^*$. In this paper, we investigate the widely adopted model-based DRO framework (Shapiro, 2017; Levy et al., 2020), where the reference distribution $P_t = P^*$ is known, and the uncertainty set radius is fixed. This enables our method to produce solutions that remain robust even when the test-time context distributions change. While the setting with the known $P^*$ is standard in the DRO literature (Shapiro, 2017; Levy et al., 2020), our framework can also apply to the scenario where the reference distribution is empirically estimated. In this case, as the estimator converges to the true reference distribution, the resulting solution remains robust with respect to the reference distribution as well.

# 3 DISTRIBUTIONALLY ROBUST MULTI-OBJECTIVE BAYESIAN OPTIMIZATION

In this section, we study the multi-objective BO problem under the general $\varphi$-divergence DRO setting. For $\varphi$-divergence DRO problems, the distance function is defined as

$$D_\varphi(Q, P_t) := \mathbb{E}_{c \sim P_t} \left[ \varphi \left( \frac{dQ}{dP_t}(c) \right) \right], \tag{6}$$

where $\varphi : \mathbb{R} \to (-\infty, \infty]$ is a convex, lower semi-continuous function with $\varphi(1) = 0$, and $\frac{dQ}{dP_t}$ is the Radon-Nikodym derivative if $Q \ll P_t$. Otherwise, we have that $D_\varphi(P, Q) = +\infty$.

For our multi-objective setting, $\boldsymbol{f}(x, c) = [f_1(x, c), \dots, f_K(x, c)]$ is a vector instead of a scalar. As mentioned in Sec 2.2, the max–min robustness defined in equation 3 is challenging to generalize to the multi-objective setting. In the MOO setting, for a fixed $x$, directly optimizing the objectives w.r.t. the unknown distribution $Q$ by the following formulation

$$\inf_{Q \in \mathcal{U}(P_t)} \mathbb{E}_{c \sim Q}[\boldsymbol{f}(x, c)], \tag{7}$$

yields the Pareto Frontier of the set $\boldsymbol{f}_\mathcal{U}(x)$ defined in equation 4. Compared with the single-objective setting, where the result of the minimization over the context distribution $Q$ is a scalar, in our MOO setting, it is not straightforward to find the optimal $x$ for the objective shown in equation 7.

To solve this problem, in this paper, we focus on the max–min robustness proposed in Ehrgott et al. (2014), which aims to find the robust efficient points defined in Definition 1. Moreover, by Theorem 4.3 of Ehrgott et al. (2014), the solutions of equation 8 are guaranteed to be robust efficient:

$$\sup_{x \in \mathcal{X}} \inf_{Q \in \mathcal{U}(P_t)} \mathbb{E}_{c \sim Q}[\boldsymbol{s}^\top \boldsymbol{f}(x, c)], \tag{8}$$

where $\Lambda := \left\{ \boldsymbol{s} | \sum_{i=1}^K s_i = 1, \text{ and } s_i > 0, \forall i \in [K] \right\}$ and $\boldsymbol{s} \in \Lambda$. Though this linear scalarization can not guarante to enumerate the robust efficient solutions, we adopt it in this paper for the following reasons: (1) This scalarization changes the objective from a vector to a scalar, making it easier to optimize. (2) The solutions obtained by this linear scalarization are guaranteed to be robust efficient.

However, it is still challenging to solve the primal formulation of the DRO problem defined in equation 8, which requires addressing a max-min problem and optimizing the expected loss under an unknown distribution. For the $\varphi$-divergence DRO problem, due to the strong duality, solving equation 8 is equivalent to solving the following dual problem (Ben-Tal et al., 2013):

$$\sup_{x \in \mathcal{X}, \lambda \geq 0, \eta \in \mathbb{R}} \left( -\eta - \lambda \varepsilon_t - \lambda \mathbb{E}_{c \sim P_t} \left[ \varphi^\star \left( \frac{-\boldsymbol{s}^\top \boldsymbol{f}(x, c) - \eta}{\lambda} \right) \right] \right), \tag{9}$$

where $\varphi^\star(u) := \sup_{u'}(uu' - \varphi(u'))$ is the conjugate function of $\varphi$. Compared with the primal formulation in equation 8, equation 9 is a simpler maximization problem, and the function is optimized under the known reference distribution $P_t$. In this paper, we consider the general $\varphi$-divergence DRO, and we provide several widely used $\varphi$-divergences in the following examples, where closed-form solutions for the dual variable $\lambda$ or $\eta$ exist. Note that in Shapiro (2017); Duchi & Namkoong (2021), only the dual formulation of $\inf_{x \in \mathcal{X}} \sup_{Q \in \mathcal{U}(P_t)} \mathbb{E}_{c \sim Q}[\boldsymbol{s}^\top \boldsymbol{f}(x, c)]$ is provided, and one can get the dual for equation 8 by studying $-\inf_{x \in \mathcal{X}} \sup_{Q \in \mathcal{U}(P_t)} -\mathbb{E}_{c \sim Q}[\boldsymbol{s}^\top \boldsymbol{f}(x, c)]$.

---

**Algorithm 1** DRMOBO for general $\varphi$-divergence

---

**Input:** Number of iterations $T$, initial dataset $D_0$
**for** $t = 1$ **to** $T$ **do**
    Sample a context $c_t \sim P_t$
    Observe $\boldsymbol{s_t} \sim P_{\boldsymbol{s}}$
    Construct the GP model given $D_{t-1}$
    Compute $x_t \in \arg\sup \alpha_t(x, \boldsymbol{s_t})$ based on equation 10
    Obtain the black-box value $y_t^i(c_t) = f_i(x_t, c_t) + \eta_t^i(c_t)$ for $i \in [K]$
    Augment the dataset $D_t = D_{t-1} \cup_{i \in [K]} (x_t, y_t^i, c_t)$
**end for**

---

**Example 1** (CVaR (Shapiro, 2017)). *For any $\alpha \in (0, 1]$, the $\varphi$ function is set to $\varphi(a) = 0$ if $a \in [0, \alpha^{-1}]$; otherwise $\varphi = +\infty$. The corresponding conjugate function is $\varphi^*(a) = [\alpha^{-1}a]_+$, where $[a]_+ = \max(a, 0)$. Then solving equation 8 is equivalent to solving*

$$\sup_{\eta \in \mathbb{R}} -\eta - \alpha^{-1} \mathbb{E}_{c \sim P_t}[(-\boldsymbol{s}^\top \boldsymbol{f}(x, c) - \eta)_+].$$

**Example 2** (TV-divergence (Shapiro, 2017)). *The $\varphi$ function is set to $\varphi(a) = |a - 1|$ if $a \geq 0$, otherwise $\varphi = +\infty$. The corresponding conjugate function is $\varphi^*(a) = [a + 1]_+ - 1$ if $a \leq 1$; otherwise $\varphi^*(a) = +\infty$. Then solving equation 8 is equivalent to solving*

$$\frac{\epsilon_t}{2} \inf_c f(x, c) + \left(1 - \frac{\epsilon_t}{2}\right) \sup_{\eta \in \mathbb{R}} \left(-\eta - \left(1 - \frac{\epsilon_t}{2}\right)^{-1} \mathbb{E}_{c \sim P_t}[(-\boldsymbol{s}^\top \boldsymbol{f}(x, c)) - \eta)_+]\right).$$

**Example 3** (Cressie-Read family (Duchi & Namkoong, 2021)). *For $k \in (1, +\infty)$, the $\varphi$ function is set to $\varphi_k(a) = \frac{a^k - ak + k - 1}{k(k-1)}$ if $a \geq 0$, otherwise $\varphi = +\infty$. The corresponding conjugate function is $\varphi^*(a) = \frac{1}{k}[((k-1)a + 1)_+^{k^*} - 1]$, where $k^* = \frac{k}{k-1}$. Then solving equation 8 is equivalent to solving*

$$\sup_{\eta \in \mathbb{R}} -\eta - (1 + k(k-1)\epsilon_t)^{\frac{1}{k}} \mathbb{E}_{c \sim P_t}[(-\boldsymbol{s}^\top \boldsymbol{f}(x, c) - \eta)_+^{k^*}]^{\frac{1}{k^*}}.$$

*Note that when $k = 2$, it reduces to the famous $\chi^2$-divergence DRO problem, whereas $k \to 1$, it corresponds to the KL-divergence DRO problem.*

**Example 4** (KL-divergence (Shapiro, 2017)). *The $\varphi$ function is set to $\varphi(a) = x \ln(x) - x + 1$ if $a \geq 0$, otherwise $\varphi = +\infty$. The corresponding conjugate function is $\varphi^*(a) = e^a - 1$. Then solving equation 8 is equivalent to solving*

$$\sup_{\lambda \geq 0} -\lambda \epsilon_t - \lambda \mathbb{E}_{c \sim P_t} \left[\exp\left(\frac{-\boldsymbol{s}^\top \boldsymbol{f}(x, c)}{\lambda}\right)\right].$$

We then introduce our method, as shown in Algorithm 1. In the MOO setting, we begin by randomly sampling the coefficients $\boldsymbol{s_t}$ based on a known distribution $P_{\boldsymbol{s}}$ with the support $\Lambda$ (Paria et al., 2020). The solution of equation 8 for each $\boldsymbol{s} \in \Lambda$ is guaranteed to be robust efficient, while the distribution $P_{\boldsymbol{s}}$ serves only to assign probability density across the resulting solutions. In this paper, we take $P_{\boldsymbol{s}}$ to be the uniform distribution. After the random scalarization, an acquisition function is defined as follows:

$$\alpha_t(x, \boldsymbol{s_t}) := \sup_{\lambda \geq 0, \eta \in \mathbb{R}} \left(-\eta - \lambda \varepsilon_t - \lambda \mathbb{E}_{c \sim P_t}\left[\varphi^\star \left(\frac{-\boldsymbol{s_t}^\top(\boldsymbol{\mu_{t-1}}(x, c) + \beta_t^{\frac{1}{2}} \boldsymbol{\sigma_{t-1}}(x, c)) - \eta}{\lambda}\right)\right]\right), \tag{10}$$

where $\boldsymbol{\mu_{t-1}}(x, c) = [\mu_{t-1}^1(x, c), \ldots, \mu_{t-1}^K(x, c)]^\top$ and $\boldsymbol{\sigma_{t-1}}(x, c) = [\sigma_{t-1}^1(x, c), \ldots, \sigma_{t-1}^K(x, c)]^\top$. We then select the action $x_t \in \arg\sup_{x \in \mathcal{X}} \alpha(x, \boldsymbol{s_t})$ that maximizes the acquisition function and sample the corresponding observation. In practice, since $\varphi^*$ is convex and both $\boldsymbol{\mu_{t-1}}$ and $\boldsymbol{\sigma_{t-1}}$ are available and both $\lambda, \eta$ are scalars, the optimal solution of equation 10 can be efficiently computed.

## 4 THEORETICAL RESULTS

Define $x_t^* \in \arg\sup_{x \in \mathcal{X}} \inf_{Q \in \mathcal{U}(P_t)} \mathbb{E}_{c \sim Q}[\boldsymbol{s}_t^\top \boldsymbol{f}(x, c)]$, which is the robust efficient solution we aim to obtain. Since $\boldsymbol{f}(x, c)$ is a black-box function that is expensive to evaluate, we then use $\boldsymbol{\mu_{t-1}}(x, c) + \beta_t^{\frac{1}{2}} \boldsymbol{\sigma_{t-1}}(x, c)$ to estimate the black-box function in our acquisition function shown in equation 10. This estimate inevitably introduces some errors, which we quantify through the regret defined as follows:

$$r_t = \inf_{Q \in \mathcal{U}(P_t)} \mathbb{E}_{c \sim Q}[\boldsymbol{s}_t^\top \boldsymbol{f}(x_t^*, c)] - \inf_{Q \in \mathcal{U}(P_t)} \mathbb{E}_{c \sim Q}[\boldsymbol{s}_t^\top \boldsymbol{f}(x_t, c)]. \tag{11}$$

This regret is widely studied (Husain et al., 2023; Kirschner et al., 2020) in the single-objective setting, and equation 11 extends it to the multi-objective setting.

To provide a theoretical analysis of the regret, we adopt the following widely used assumptions:

**Assumption 1.** *Let $\mathcal{X} \subseteq [-r, r]^d$ be a compact and convex set (Paria et al., 2020; Srinivas et al., 2009), where $d \in \mathbb{N}$ and $r > 0$. The underlying distribution of the context $P^*$ is discrete and $|\mathcal{C}|$ is bounded.*

**Assumption 2.** *We assume that for each $i \in [K]$, the kernel $k_i((x, c), (x', c'))$ meets the requirements (Paria et al., 2020; Srinivas et al., 2009) such that the derivatives of $f_i$ are upper bounded with high probability. To be more detailed, there exist some constants $a, b > 0$ such that for any $c \in \mathcal{C}$ and $j \in [d]$,*

$$\sup_{x \in \mathcal{X}} \left| \frac{\partial f_i}{\partial x_j} \right| > L \tag{12}$$

*holds with probability not larger than $ae^{-(L/b)^2}$.*

Assumption 2 s a standard smoothness condition in BO and requires that the objective functions do not vary too fast with respect to $x$. The theoretical results in Husain et al. (2023); Tay et al. (2022) require that $\sup_{(x,c) \in \mathcal{X} \times \mathcal{C}} f(x, c)$ or the RKHS norm $\|f\|_k$ is bounded in the single-objective setting. In contrast, our analysis does not rely on such assumptions, yet we establish regret bounds with the same convergence rate through a tighter analysis. In fact, $f(x, c) \sim GP(m, k)$ and the function value of $f$ can be unbounded. Our regret bound depends on the maximum information gain $\gamma_T^i := \sup_{A_T} I(\boldsymbol{y}_T^i; \boldsymbol{f}_T^i)$ for each objective $i$, where $I$ is the mutual information, $\boldsymbol{y}_t^i = [y_1^i, ..., y_t^i]^\top$, $\boldsymbol{f}_t^i = [f_1^i, ..., f_T^i]^\top$ and $A_T = \{(x_1, c_1), ..., (x_t, c_T)\} \subset \mathcal{X} \times \mathcal{C}$ is the dataset. Moreover, with the RBF kernel (or Matern kernel with $\nu > 1$), it can be shown that $\gamma_T^i \leq \mathcal{O}(\log(T)^{d+d'+1})$ $\left( \text{or } \gamma_T^i \leq \mathcal{O}(T^{\frac{(d+d')(d+d'+1)}{2\nu+(d+d')(d+d'+1)}} \log(T)) \right)$ (Husain et al., 2023). We assume that there exists an $M > 0$ such that $M = \sup\{m : \varphi(m) \leq \frac{\epsilon_t}{\inf_c P_t(c)}\}$. This condition is easily verified to hold for all of our examples. Under this assumption, for any $Q \in \mathcal{U}(P_t), c \in \mathcal{C}$, it can be shown that $\frac{Q(c)}{P_t(c)} \leq M$. Otherwise we have that $D_\varphi(Q, P_t) = \sum_{c \in \mathcal{C}} P_t(c) \varphi \left( \frac{Q(c)}{P_t(c)} \right) > \varphi(m) \inf_c P_t(c) \geq \epsilon_t$, which contradicts $Q \in \mathcal{U}(P_t)$. Then, by constructing a finite $\epsilon$-net $\mathcal{X}_t$ at time $t$, we have that:

**Theorem 1.** *With Assumptions 1 and 2 holding for each $f_i$, let $|\mathcal{X}_t| = (2dt^2br\sqrt{\log(8d|\mathcal{C}|a/\delta)})^d$, $\beta_t = 2\log(8|\mathcal{X}_t||\mathcal{C}|\pi_t/\delta)$ for any $0 < \delta < 1$ and $\pi_t = \frac{\pi^2 t^2}{6}$, it follows that*

$$\sum_{t=1}^{T} r_t \leq 4M\sqrt{\sum_{i=1}^{K} \frac{2\beta_T \gamma_T^i T}{\log(1 + \sigma_f^{-2})}} + 16MK\log\left(\frac{24K}{\delta}\right) + \frac{\pi^2}{6}$$

*holds with the probability larger than $1 - \delta$.*

Our proposed algorithm and theoretical results are directly applicable to the single-objective setting (see Appendix B). While most existing theoretical analyses focus only on single-objective formulations, our results have several advantages even in the single-objective setting: Compared with Srinivas et al. (2009), which does not consider the robustness, our solution achieves the robustness at the cost of only an additional term $16MK\log\left(\frac{24K}{\delta}\right)$. Compared with the robust BO method (Huang et al., 2024), where the regret bound is of order $\mathcal{O}(\sqrt{\sum_{t=1}^{T} \epsilon_t^2} + T^{\frac{2+D_c}{4+D_c}})$ with $D_c$ denoting

the dimension of the context, our regret bound depends only logarithmically on $|\mathcal{C}|$ without requiring the uncertainty set radius to shrink to guarantee sublinear regret.

Beyond regret guarantees, our method advances robust BO by relaxing assumptions and broadening applicability. In particular, it eliminates the bounded function value assumption (Husain et al., 2023; Tay et al., 2022) and supports a broad class of $\varphi$-divergence DRO uncertainty sets without requiring the uncertainty set radius to shrink in order to guarantee a sublinear regret. To achieve that, we first construct an $\epsilon$-net over the general set $\mathcal{X}$ and then derive a tighter upper bound on the estimation error between the black-box function and its surrogate. We then show the proof sketch.

*Proof Sketch.* The detailed proof can be found in Appendix B.3. Note that the set $\mathcal{X}$ can be continuous and we construct a finite $\epsilon$-net $\mathcal{X}_t$ of size $(2\tau_t)^d$ to guarantee that for all $x \in \mathcal{X}$, $\|x - [x]_t\|_1 \le rd/\tau_t$, where $[x]_t$ represents the closest point in the net $\mathcal{X}_t$. In this paper, we construct a net that has $2\tau_t$ points uniformly distributed in each coordinate. We then have that:

**Lemma 1.** *For any $\delta \in (0, 1)$, let $|\mathcal{X}_t| = (2dt^2 br\sqrt{\log(8d|\mathcal{C}|a/\delta)})^d$ and $\beta_t = 2\log(8|\mathcal{X}_t||\mathcal{C}|\pi_t/\delta)$. For $\pi_t > 0$ such that $\sum_{t \ge 1} \pi_t^{-1} = 1$, we have that*

$$\forall t \ge 0, \forall x \in \mathcal{X}, \forall c \in C, |f_i(x, c) - \mu_{t-1}^i([x]_t, c)| \le \beta_t^{\frac{1}{2}} \sigma_{t-1}^i([x]_t, c) + \frac{1}{t^2}$$

*holds for each $i \in [K]$ with probability at least $1 - \frac{\delta}{4}$.*

By Lemma 1, each black-box function $f_i$ can be decomposed into two components: the surrogate estimate $\mu_{t-1}^i + \beta_t^{\frac{1}{2}} \sigma_{t-1}^i$ and the corresponding estimation error. Due to the fact that $x_t$ maximizes the acquisition function defined in equation 10 and the estimate is an upper bound on the true worst-case performance, we can further show that

$$r_t \le 2 \sum_{i=1}^K \mathbb{E}_{c \sim Q_t^*}[\beta_t^{\frac{1}{2}} \sigma_{t-1}^i(x_t, c)] + \text{other terms},$$

where $Q_t^* \in \arg\inf_{Q \in \mathcal{U}(P_t)} \mathbb{E}_{c \sim Q}[\boldsymbol{s}_t^\top \boldsymbol{f}(x_t, c)]$. In the Bayesian optimization setting with $|\mathcal{C}| = 1$, $\sum_{t=1}^T \mathbb{E}_{c \sim Q_t^*}[\beta_t^{\frac{1}{2}} \sigma_{t-1}^i(x_t, c)]$ reduces to $\sum_{t=1}^T \beta_t^{\frac{1}{2}} \sigma_{t-1}^i(x_t)$, which can be upper bounded by a function of the maximum information gain: $\sqrt{\frac{2T\beta_T \gamma_T^i}{\log(1 + \sigma_f^{-2})}}$ with non-decreasing $\beta_t$ according to the Cauchy-Schwarz inequality and the following Lemma 2.

**Lemma 2.** *(Information bound in Srinivas et al. (2009)) The sum of predictive variances can be upper bounded by the information gain as follows:*

$$\sum_{t=1}^T \sigma_{t-1}^i(x_t, c_t)^2 \le \frac{2\gamma_T^i}{\log(1 + \sigma_f^{-2})}.$$

However, in DRO setting, the LHS becomes $\sum_{t=1}^T \mathbb{E}_{c \sim Q_t^*}\left[\beta_t^{1/2} \sigma_{t-1}^i(x_t, c)\right]$, which is an expectation under $Q_t^*$ and the information bound in Lemma 2 only applies to $\sum_{t=1}^T \beta_t^{1/2} \sigma_{t-1}^i(x_t, c_t)$. Therefore, Lemma 2 cannot be invoked directly, and additional analysis is needed.

Since $Q_t^* \in \mathcal{U}(P_t)$, it follows that $\frac{Q_t^*(c)}{P_t(c)} \le M$ for all $c \in \mathcal{C}$. Thus, the expectation under $Q_t^*$ can be bounded by at most a constant factor $M$ times the corresponding expectation under $P_t$. This reduction allows us to control $\sum_{t=1}^T \mathbb{E}_{c \sim Q_t^*}\left[\beta_t^{1/2} \sigma_{t-1}^i(x_t, c)\right]$ via the simpler form $\sum_{t=1}^T \beta_t^{1/2} \sigma_{t-1}^i(x_t, c_t)$, evaluated only at the sampled contexts $c_t$. As a result, the information bound in Lemma 2 can be applied, completing the proof. □

## 5 NUMERICAL RESULTS

We then present numerical experiments for the multi-objective setting, conducted on synthetic functions. More details and further experiments are provided in Appendix D. We evaluate our method

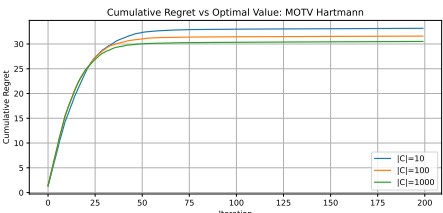
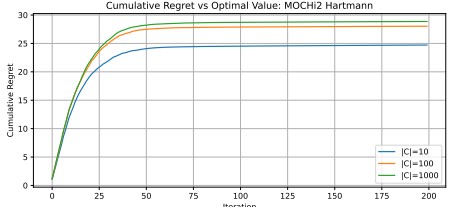

(a) Regret vs Iteration for Hartmann with TV DRO    (b) Regret vs Iteration for Hartmann with $\chi^2$-DRO

Figure 1: Results for multi-objective Hartmann functions.

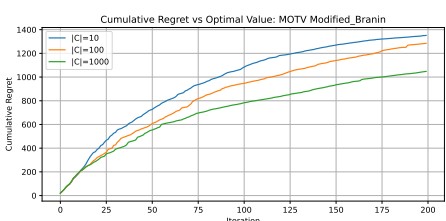
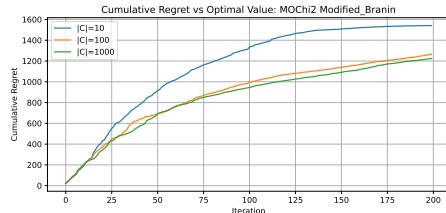

(a) Regret vs Iteration for Modified Branin with TV DRO    (b) Regret vs Iteration for Modified Branin with $\chi^2$-DRO

Figure 2: Results for multi-objective Modified Branin functions.

on three widely used synthetic functions with context variables (Surjanovic & Bingham), e.g., the Ackley function, Modified Branin function, and Hartmann function. In our multi-objective formulation, all objectives share a common context variable $c$. Despite Huang et al. (2024) only focusing on the single-objective setting, while we study the multi-objective setting, additional key differences distinguish the problem studied in this paper: **1. Context environment:** Huang et al. (2024) choose a Gaussian kernel to estimate the context distribution, and in this paper, we follow the method in Huang et al. (2024); Kirschner et al. (2020) to discretize the context space into the size of $\lceil |\tilde{\mathcal{C}}|^{1/D} \rceil^D$, where $D$ is the dimension of the context variable; **2. Model-based setting:** While Huang et al. (2024) assume a decreasing radius of the DRO uncertainty set, we instead consider a constant radius to obtain solutions that remain robust under shifted context distributions; and **3. Acquisition function**: From an algorithmic standpoint, the primary distinction between Huang et al. (2024) and this work lies in the acquisition function, which corresponds to the dual formulation of the DRO problem. In this paper, we investigate the model-based setting with a constant DRO radius and further extend the TV-DRO problem to a broader class of $\varphi$-divergence problems. Our results for the Modified Branin and Hartmann functions under both TV- and $\chi^2$-DRO settings are presented in Figures 1 and 2. The results show that, even with a fixed radius of the DRO uncertainty sets, the cumulative regret exhibits sublinear growth, which is consistent with our theoretical results.

## 6 CONCLUSION

In this paper, we investigate the Bayesian optimization problem in contextual environments, where we consider the distributional robustness in the context variable. We develop frameworks for both single- and multi-objective settings and establish provable sublinear regrets. Compared with existing works under the single-objective setting, our method does not require a decaying radius of DRO uncertainty sets and applies to a wide range of $\varphi$-divergence DRO problems. For the multi-objective setting, our method can obtain robust efficient solutions (Ehrgott et al., 2014), with regret bounds of the same order as in the single-objective case. We further validate the effectiveness of our approach through numerical experiments. In future work, we aim to extend our framework to support additional DRO uncertainty sets, such as the MMD and Wasserstein sets. We also plan to apply the framework to large-scale real-world applications, such as vehicle design and drug discovery.

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

---

**Algorithm 2** DRBO for general $\varphi$-divergence

---

**Input:** Number of iterations $T$, initial dataset $D_0$
**for** $t = 1$ **to** $T$ **do**
    Sample a context $c_t \sim P^*$
    Construct the GP model given $D_{t-1}$
    Compute $x_t \in \arg \sup \alpha_t(x)$ based on equation 14
    Obtain the black-box value $y_t = f(x_t, c_t) + \eta_t$
    Augment the dataset $D_t = D_{t-1} \cup (x_t, y_t, c_t)$
**end for**

---

## A    USE OF LARGE LANGUAGE MODELS

In this paper, we only use LLMs to aid and polish writing.

## B    SINGLE OBJECTIVE

**Single-objective model:** We study a black-box reward function $f : \mathcal{X} \times \mathcal{C} \to \mathbb{R}$, where $x \in \mathcal{X} \subseteq \mathbb{R}^d$ is the model parameter and $c \in \mathcal{C} \subseteq \mathbb{R}^{d'}$ is the context variable. We assume at time $t$, the context variable is $c_t$, and given model parameter $x_t$, we obtain the current observation $y_t = f(x_t, c_t) + \eta_t$, where $\eta_t \sim \mathcal{N}(0, \sigma_f^2)$ is the noise and $\sigma_f^2$ is the constant noise variance. The following GP (Williams & Rasmussen, 2006; Krause & Ong, 2011; van Bueren et al., 2021) assumption is widely used in the BO literature: $f \sim GP(m, k)$, where $m : \mathcal{X} \times \mathcal{C} \to \mathbb{R}$ is a known mean function and $k : \mathcal{X} \times \mathcal{C} \times \mathcal{X} \times \mathcal{C} \to \mathbb{R}$ is a known covariance function. We follow the widely used assumptions that $m \equiv 0$ and $k((x, c), (x, c)) \leq 1$ for any $x \in \mathcal{X}$ and $c \in \mathcal{C}$. Denote by $A_t = \{(x_1, c_1), ..., (x_t, c_t)\}$ the dataset at time $t$, and let $\boldsymbol{y_t} = [y_1, ..., y_t]^\top$ be the observation vector. Given $A_t$ and $\boldsymbol{y_t}$, it can be shown that the posterior over $f(x, c)$ follows a GP distribution, whose mean and variance are given by

$$\mu_t(x, c) = \boldsymbol{k_t}(x, c)^\top (\boldsymbol{K_t} + \sigma_f^2 \boldsymbol{I})^{-1} \boldsymbol{y_t},$$
$$(\sigma_t(x, c))^2 = k([x, c], [x, c]) - \boldsymbol{k_t}(x, c)^\top (\boldsymbol{K_t} + \sigma_f^2 \boldsymbol{I})^{-1} \boldsymbol{k_t}(x, c), \tag{13}$$

where we define the kernel vector $\boldsymbol{k_t}([x, c]) = [k([x_1, c_1], [x, c]), ..., k([x_t, c_t], [x, c])]^\top$ and the positive definite kernel matrix $\boldsymbol{K_t} = [k([x_a, c_a], [x_b, c_b])]_{a,b \leq t}$.

We then provide our method, shown in Algorithm 2. We first sample a context based on the underlying distribution $P_t$. According to equation 13, given $x_t$ and $c_t$, we can estimate the mean and variance of $f(x_t, c_t)$. For the general $\varphi$-divergence DRO problem, we define the acquisition function as follows:

$$\alpha_t(x) := \sup_{\lambda \geq 0, \eta \in \mathbb{R}} \left( -\eta - \lambda \varepsilon_t - \lambda \mathbb{E}_{c \sim P_t} \left[ \varphi^\star \left( \frac{-\mu_{t-1}(x, c) - \beta_t^{\frac{1}{2}} \sigma_{t-1}(x, c) - \eta}{\lambda} \right) \right] \right), \tag{14}$$

where $\mu_{t-1}(x, c) + \beta_t^{\frac{1}{2}} \sigma_{t-1}(x, c)$ is used to estimate $f(x, c)$. At time $t$, the model parameter $x_t \in \arg \sup_{x \in \mathcal{X}} \alpha_t(x)$ is selected to maximize the acquisition function. It is hard to solve the max–min problem in equation 3 directly. However, maximizing the acquisition function defined in equation 14 is easy since the distribution is known and it is a pure maximization problem.

The following regret (Husain et al., 2023; Kirschner et al., 2020) is defined to measure the performance gap between our selected $x_t$ and the optimal $x_t^*$ such that $x_t^* \in \arg \sup_{x \in \mathcal{X}} \inf_{Q \in \mathcal{U}(P_t)} \mathbb{E}_{c \sim Q}[f(x, c)]$:

$$r_t = \inf_{Q \in \mathcal{U}(P_t)} \mathbb{E}_{c \sim Q}[f(x_t^*, c)] - \inf_{Q \in \mathcal{U}(P_t)} \mathbb{E}_{c \sim Q}[f(x_t, c)]. \tag{15}$$

Our regret bound depends on the maximum information gain $\gamma_T := \sup_{A_T} I(\boldsymbol{y_T}; \boldsymbol{f_T})$ at time $T$, where $I$ is the mutual information, $\boldsymbol{y_t} = [y_1, ..., y_t]^\top$, $\boldsymbol{f_t} = [f_1, ..., f_T]^\top$ and $A_T = \{(x_1, c_1), ..., (x_t, c_T)\} \subset \mathcal{X} \times \mathcal{C}$ is the dataset. Then, we have the following theorem:

**Theorem 2.** *Under Assumptions 1 and 2, let $\beta_t = 2\log(8|\mathcal{X}_t||\mathcal{C}|\pi_t/\delta)$ for $\pi_t = \frac{\pi^2 t^2}{6}$, and $|\mathcal{X}_t| = (2dt^2 br\sqrt{\log(8d|\mathcal{C}|a/\delta)})^d$. It then follows that*

$$\sum_{t=1}^{T} r_t \leq 4M\sqrt{\frac{2\beta_T \gamma_T T}{\log(1 + \sigma_f^{-2})}} + 16M\log\left(\frac{24}{\delta}\right) + \frac{\pi^2}{6}$$

*holds with the probability at least $1 - \delta$.*

### B.1 PROOF OF LEMMA 1

*Proof.* By assumption 2 and the union bound, we can show that

$$\Pr\{\forall j, \forall x \in \mathcal{X}, \text{and } \forall c \in \mathcal{C}, |\partial f/(\partial x_j)| < L\} \geq 1 - d|\mathcal{C}|ae^{-L^2/b^2}.$$

It follows that

$$\forall x, x' \in \mathcal{X}, \text{and } c \in \mathcal{C}, \ |f(x, c) - f(x', c)| \leq L\|x - x'\|_1$$

holds with the probability at least $1 - d|\mathcal{C}|ae^{-L^2/b^2}$. Set $L = b\sqrt{\log(8d|\mathcal{C}|a/\delta)}$ and $\tau_t = dt^2 br\sqrt{\log(8d|\mathcal{C}|a/\delta)}$, we have that

$$\forall x \in \mathcal{X}, \quad |f(x) - f([x]_t)| \leq b\sqrt{\log(8d|\mathcal{C}|a/\delta)}\|x - [x]_t\|_1 \leq \frac{1}{t^2} \tag{16}$$

holds with probability at least $1 - \frac{\delta}{8}$. With $\tau_t = dt^2 br\sqrt{\log(8d|\mathcal{C}|a/\delta)}$, it can be shown that $|\mathcal{X}_t| = (2dt^2 br\sqrt{\log(8d|\mathcal{C}|a/\delta)})^d$. Based on Lemma 3 and the union bound, setting $\beta_t = 2\log(8|\mathcal{X}_t||\mathcal{C}|\pi_t/\delta)$, we can show that

$$\forall x \in \mathcal{X}_t, \forall t \geq 1 \text{ and } \forall c \in \mathcal{C}, |f(x, c) - \mu_{t-1}(x, c)| \leq \beta_t^{\frac{1}{2}}\sigma_{t-1}(x, c) \tag{17}$$

holds with probability $\geq 1 - \frac{\delta}{8}$. Based on equation 16, equation 17, and the union bound, it can be further shown that

$$\forall x \in \mathcal{X}, \forall t \geq 1 \text{ and } \forall c \in \mathcal{C}, |f(x, c) - \mu_{t-1}([x]_t, c)| \leq \beta_t^{\frac{1}{2}}\sigma_{t-1}([x]_t, c) + \frac{1}{t^2}$$

holds with probability $\geq 1 - \frac{\delta}{4}$. This completes the proof. $\square$

**Lemma 3.** *(Adapted Lemma 5.6 in Srinivas et al. (2009) ) For any $0 < \delta < 1$, set $\beta_t = 2\log(|\mathcal{X}_t|\pi_t/\delta)$ with $\sum_{t \geq 1} \pi_t^{-1} = 1$ and $\pi_t > 0$. Then, for any $c \in \mathcal{C}$, it can be shown that*

$$\forall x \in \mathcal{X}_t, \forall t \geq 1, |f(x, c) - \mu_{t-1}(x, c)| \leq \beta_t^{\frac{1}{2}}\sigma_{t-1}(x, c)$$

*holds with probability at least $1 - \delta$.*

### B.2 DETAILS ABOUT LEMMA 2

Denote by the maximum information gain $\gamma_T := \sup_{A_T} I(\boldsymbol{y}_T; \boldsymbol{f}_T)$ at time $T$, where $I$ is the mutual information, $\boldsymbol{y}_t = [y_1, ..., y_t]^\top$, $\boldsymbol{f}_t = [f_1, ..., f_T]^\top$ and $A_T = \{(x_1, c_1), ..., (x_t, c_T)\} \subset \mathcal{X} \times \mathcal{C}$ is the dataset. Based on Lemma 5.3 and Lemma 5.4 in Srinivas et al. (2009), we have the following lemma:

**Lemma 4.** *(Information bound in Srinivas et al. (2009)) The sum of predictive variances can be upper bounded by the information gain as follows:*

$$\sum_{t=1}^{T} \sigma_{t-1}^2(x_t, c_t) \leq \frac{2\gamma_T}{\log(1 + \sigma_f^{-2})}.$$

### B.3 PROOF OF THEOREM 2

*Proof.* The regret can be written as

$$r_t = \inf_{Q \in \mathcal{U}(P_t)} \mathbb{E}_{c \sim Q}[f(x_t^*, c)] - \inf_{Q \in \mathcal{U}(P_t)} \mathbb{E}_{c \sim Q}[f(x_t, c)],$$

which can be further bounded as

$$r_t = \inf_{Q \in \mathcal{U}(P_t)} \mathbb{E}_{c \sim Q}[f(x_t^*, c)] - \inf_{Q \in \mathcal{U}(P_t)} \mathbb{E}_{c \sim Q}[f(x_t, c)]$$

$$= \inf_{Q \in \mathcal{U}(P_t)} \mathbb{E}_{c \sim Q}[f(x_t^*, c)] - \inf_{Q \in \mathcal{U}(P_t)} \mathbb{E}_{c \sim Q}[\mu_{t-1}([x_t^*]_t, c) + \beta_t^{\frac{1}{2}} \sigma_{t-1}([x_t^*]_t, c)]$$

$$+ \inf_{Q \in \mathcal{U}(P_t)} \mathbb{E}_{c \sim Q}[\mu_{t-1}([x_t^*]_t, c) + \beta_t^{\frac{1}{2}} \sigma_{t-1}([x_t^*]_t, c)]$$

$$- \inf_{Q \in \mathcal{U}(P_t)} \mathbb{E}_{c \sim Q}[\mu_{t-1}(x_t, c) + \beta_t^{\frac{1}{2}} \sigma_{t-1}(x_t, c)]$$

$$+ \inf_{Q \in \mathcal{U}(P_t)} \mathbb{E}_{c \sim Q}[\mu_{t-1}(x_t, c) + \beta_t^{\frac{1}{2}} \sigma_{t-1}(x_t, c)] - \inf_{Q \in \mathcal{U}(P_t)} \mathbb{E}_{c \sim Q}[f(x_t, c)]. \quad (18)$$

Based on the dual formulation, for $\varphi$-divergence DRO problem, we can show that

$$\alpha_t(x_t) = \inf_{Q \in \mathcal{U}(P_t)} \mathbb{E}_{c \sim Q}[\mu_{t-1}(x_t, c) + \beta_t^{\frac{1}{2}} \sigma_{t-1}(x_t, c)],$$

where $\alpha_t$ is defined in equation 14. Since $x_t \in \arg\sup_x \alpha_t(x)$, it can be shown that

$$\inf_{Q \in \mathcal{U}(P_t)} \mathbb{E}_{c \sim Q}[\mu_{t-1}([x_t^*]_t, c) + \beta_t^{\frac{1}{2}} \sigma_{t-1}([x_t^*]_t, c)] - \inf_{Q \in \mathcal{U}(P_t)} \mathbb{E}_{c \sim Q}[\mu_{t-1}(x_t, c) + \beta_t^{\frac{1}{2}} \sigma_{t-1}(x_t, c)]$$

$$\leq \alpha_t([x_t^*]_t) - \alpha_t(x_t) \leq 0. \quad (19)$$

According to Lemma 1, it follows that

$$|f(x_t^*, c) - \mu_{t-1}([x_t^*]_t, c)| \leq \beta_t^{\frac{1}{2}} \sigma_{t-1}([x_t^*]_t, c) + \frac{1}{t^2}, \quad \forall t \geq 1 \quad (20)$$

holds with probability $\geq 1 - \frac{\delta}{4}$. With this inequality, we can further show that

$$\inf_{Q \in \mathcal{U}(P_t)} \mathbb{E}_{c \sim Q}[f(x_t^*, c)] - \inf_{Q \in \mathcal{U}(P_t)} \mathbb{E}_{c \sim Q}[\mu_{t-1}([x_t^*]_t, c) + \beta_t^{\frac{1}{2}} \sigma_{t-1}([x_t^*]_t, c)]$$

$$\leq \inf_{Q \in \mathcal{U}(P_t)} \mathbb{E}_{c \sim Q}[f(x_t^*, c)] - \inf_{Q \in \mathcal{U}(P_t)} \mathbb{E}_{c \sim Q}\left[\mu_{t-1}([x_t^*]_t, c) + \beta_t^{\frac{1}{2}} \sigma_{t-1}([x_t^*]_t, c) + \frac{1}{t^2}\right] + \frac{1}{t^2}$$

$$\leq \frac{1}{t^2}. \quad (21)$$

We then bound the last item shown in equation 18, which can be expressed by

$$\inf_{Q \in \mathcal{U}(P_t)} \mathbb{E}_{c \sim Q}[\mu_{t-1}(x_t, c) + \beta_t^{\frac{1}{2}} \sigma_{t-1}(x_t, c)] - \inf_{Q \in \mathcal{U}(P_t)} \mathbb{E}_{c \sim Q}[f(x_t, c)]$$

$$\leq \mathbb{E}_{c \sim Q_t^*}[\mu_{t-1}(x_t, c) + \beta_t^{\frac{1}{2}} \sigma_{t-1}(x_t, c) - f(x_t, c)], \quad (22)$$

where $Q_t^* \in \arg\inf_{Q \in \mathcal{U}(P_t)} \mathbb{E}_{c \sim Q}[f(x_t, c)]$ and the inequality holds due to the fact that $\inf_{Q \in \mathcal{U}(P_t)} \mathbb{E}_{c \sim Q}[\mu_{t-1}(x_t, c) + \beta_t^{\frac{1}{2}} \sigma_{t-1}(x_t, c)] \leq \mathbb{E}_{c \sim Q_t^*}[\mu_{t-1}(x_t, c) + \beta_t^{\frac{1}{2}} \sigma_{t-1}(x_t, c)]$ and $\inf_{Q \in \mathcal{U}(P_t)} \mathbb{E}_{c \sim Q}[f(x_t, c)] = \inf_{Q \in \mathcal{U}(P_t)} \mathbb{E}_{c \sim Q}[f(x_t, c)]$. Similar to Lemma 3, with our selected $\beta_t$, it can be proved that

$$\forall c \in \mathcal{C}, \forall t \geq 1, |f(x_t, c) - \mu_{t-1}(x_t, c)| \leq \beta_t^{\frac{1}{2}} \sigma_{t-1}(x_t, c) \quad (23)$$

holds with probability at least $1 - \frac{\delta}{4}$. With the inequality in equation 23 holds, equation 22 can be further bounded as

$$\inf_{Q \in \mathcal{U}(P_t)} \mathbb{E}_{c \sim Q}[\mu_{t-1}(x_t, c) + \beta_t^{\frac{1}{2}} \sigma_{t-1}(x_t, c)] - \inf_{Q \in \mathcal{U}(P_t)} \mathbb{E}_{c \sim Q}[f(x_t, c)]$$

$$\leq \mathbb{E}_{c \sim Q_t^*}[\mu_{t-1}(x_t, c) + \beta_t^{\frac{1}{2}} \sigma_{t-1}(x_t, c) - f(x_t, c)]$$

$$\leq 2\mathbb{E}_{c \sim Q_t^*}[\beta_t^{\frac{1}{2}} \sigma_{t-1}(x_t, c)]. \quad (24)$$

Thus, based on the union bound, combining equation 18, equation 19, equation 21 and equation 22, we have that

$$r_t \leq 2\mathbb{E}_{c \sim Q_t^*}[\beta_t^{\frac{1}{2}} \sigma_{t-1}(x_t, c)] + \frac{1}{t^2} \tag{25}$$

hold with probability at least $1 - \frac{\delta}{2}$. It is hard to directly bound $\sum_{t=1}^{T} \mathbb{E}_{c \sim Q_t^*}[\beta_t^{\frac{1}{2}} \sigma_{t-1}(x_t, c)]$ based on the information gain shown in Lemma 2, which can only bound $\sum_{t=1}^{T} \sigma_{t-1}^2(x_t, c_t)$. Based on the definition of uncertainty set in equation 2 and equation 6, for any $c \in \mathcal{C}$, we have that

$$P_t(C)\varphi\left(\frac{Q_t^*(C)}{P_t(C)}\right) \leq \sum_{c \in \mathcal{C}} P_t(C)\varphi\left(\frac{Q_t^*(C)}{P_t(C)}\right) \leq \epsilon_t.$$

Thus there exists an upper bound on $M > 0$ such that $\frac{Q_t^*}{P_t} \leq M$. As a result, we have that

$$\mathbb{E}_{c \sim Q_t^*}[\beta_t^{\frac{1}{2}} \sigma_{t-1}(x_t, c)] \leq M\mathbb{E}_{c \sim P_t}[\beta_t^{\frac{1}{2}} \sigma_{t-1}(x_t, c)]. \tag{26}$$

Applying Lemma 5 to equation 26, it can be proved that

$$\sum_{t=1}^{T} M\mathbb{E}_{c \sim P_t}[\beta_t^{\frac{1}{2}} \sigma_{t-1}(x_t, c)] \leq 2M \sum_{t=1}^{T} \beta_t^{\frac{1}{2}} \sigma_{t-1}(x_t, c_t) + 8M \log\left(\frac{24}{\delta}\right) \tag{27}$$

holds with probability at least $1 - \frac{\delta}{4}$. Based on the information bound in Lemma 2, we have that

$$\sum_{t=1}^{T} \sigma_{t-1}^2(x_t, c_t) \leq \frac{2}{\log(1 + \sigma_f^{-2})} \gamma_T,$$

which further implies that

$$\sum_{t=1}^{T} \sigma_{t-1}(x_t, c_t) \leq \sqrt{\frac{2\gamma_T T}{\log(1 + \sigma_f^{-2})}}. \tag{28}$$

Combining equation 25, equation 26, equation 27 and equation 28, the fact that $\beta_t$ is increasing, and by the union bound, we can prove that

$$\sum_{t=1}^{T} r_t \leq 4M\sqrt{\frac{2\beta_T \gamma_T T}{\log(1 + \sigma_f^{-2})}} + 16M \log\left(\frac{24}{\delta}\right) + \frac{\pi^2}{6}$$

holds with probability at least $1 - \delta$. This completes the proof. $\qquad \square$

**Lemma 5.** *(Lemma 3 in Kirschner & Krause (2018).) Denote by $S_t \geq 0$ a non-negative stochastic process, which is adapted to a filtration $\{\mathcal{F}_t\}$. Let $m_t = \mathbb{E}[S_t | \mathcal{F}_{t-1}]$ be the conditional expectation. We assume that there exists a $B \geq 1$ such that $S_t \leq B$. Then we have the following inequality*

$$\sum_{t=1}^{T} m_t \leq 2 \sum_{t=1}^{T} S_t + 4B \log \frac{1}{\delta} + 8B \log(4B) + 1$$

$$\leq 2 \sum_{t=1}^{T} S_t + 8B \log \frac{6B}{\delta}$$

*holds with probability at least $1 - \delta$ for any $T \geq 1$ and $0 < \delta < 1$.*

## C  MULTI-OBJECTIVE SETTING

### C.1  PROOF OF THEOREM 1

The regret can be written as

$$r_t = \inf_{Q \in \mathcal{U}(P_t)} \mathbb{E}_{c \sim Q}[\boldsymbol{s}_t^\top \boldsymbol{f}(x_t^*, c)] - \inf_{Q \in \mathcal{U}(P_t)} \mathbb{E}_{c \sim Q}[\boldsymbol{s}_t^\top \boldsymbol{f}(x_t, c)],$$

which can be further bounded as

$$r_t = \inf_{Q \in \mathcal{U}(P_t)} \mathbb{E}_{c \sim Q}[\boldsymbol{s_t}^\top \boldsymbol{f}(x_t^*, c)] - \inf_{Q \in \mathcal{U}(P_t)} \mathbb{E}_{c \sim Q}[\boldsymbol{s_t}^\top \boldsymbol{f}(x_t, c)]$$

$$= \inf_{Q \in \mathcal{U}(P_t)} \mathbb{E}_{c \sim Q}[\boldsymbol{s_t}^\top \boldsymbol{f}(x_t^*, c)] - \inf_{Q \in \mathcal{U}(P_t)} \mathbb{E}_{c \sim Q}[\boldsymbol{s_t}^\top (\boldsymbol{\mu_{t-1}}([x_t^*]_t, c) + \beta_t^{\frac{1}{2}} \boldsymbol{\sigma_{t-1}}([x_t^*]_t, c))]$$

$$+ \inf_{Q \in \mathcal{U}(P_t)} \mathbb{E}_{c \sim Q}[\boldsymbol{s_t}^\top (\boldsymbol{\mu_{t-1}}([x_t^*]_t, c) + \beta_t^{\frac{1}{2}} \boldsymbol{\sigma_{t-1}}([x_t^*]_t, c))]$$

$$- \inf_{Q \in \mathcal{U}(P_t)} \mathbb{E}_{c \sim Q}[\boldsymbol{s_t}^\top (\boldsymbol{\mu_{t-1}}(x_t, c) + \beta_t^{\frac{1}{2}} \boldsymbol{\sigma_{t-1}}(x_t, c))]$$

$$+ \inf_{Q \in \mathcal{U}(P_t)} \mathbb{E}_{c \sim Q}[\boldsymbol{s_t}^\top (\boldsymbol{\mu_{t-1}}(x_t, c) + \beta_t^{\frac{1}{2}} \boldsymbol{\sigma_{t-1}}(x_t, c))] - \inf_{Q \in \mathcal{U}(P_t)} \mathbb{E}_{c \sim Q}[\boldsymbol{s_t}^\top \boldsymbol{f}(x_t, c)]. \quad (29)$$

Based on the dual formulation, for $\varphi$-divergence DRO problem, we can show that

$$\alpha_t(x_t, \boldsymbol{s_t}) = \inf_{Q \in \mathcal{U}(P_t)} \mathbb{E}_{c \sim Q}[\boldsymbol{s_t}^\top (\boldsymbol{\mu_{t-1}}(x_t, c) + \beta_t^{\frac{1}{2}} \boldsymbol{\sigma_{t-1}}(x_t, c))],$$

where $\alpha_t(x, \boldsymbol{s})$ is defined in equation 10. Since $x_t \in \arg\sup_x \alpha_t(x, \boldsymbol{s_t})$, it can be shown that

$$\inf_{Q \in \mathcal{U}(P_t)} \mathbb{E}_{c \sim Q}[\boldsymbol{s_t}^\top (\boldsymbol{\mu_{t-1}}([x_t^*]_t, c) + \beta_t^{\frac{1}{2}} \boldsymbol{\sigma_{t-1}}([x_t^*]_t, c))]$$

$$- \inf_{Q \in \mathcal{U}(P_t)} \mathbb{E}_{c \sim Q}[\boldsymbol{s_t}^\top (\boldsymbol{\mu_{t-1}}(x_t, c) + \beta_t^{\frac{1}{2}} \boldsymbol{\sigma_{t-1}}(x_t, c))]$$

$$\leq \alpha_t([x_t^*]_t) - \alpha_t(x_t) \leq 0. \quad (30)$$

We can then slightly change Lemma 1. By setting $\beta_t = 2\log(8K|\mathcal{X}_t||\mathcal{C}|\pi_t/\delta)$ for $\sum_{t \geq 1} \pi_t^{-1} = 1$, it can be shown that for any $i \in [K]$,

$$\forall t \geq 1, \quad |f_i(x_t^*, c) - \mu_{t-1}^i([x_t^*]_t, c)| \leq \beta_t^{\frac{1}{2}} \sigma_{t-1}^i([x_t^*]_t, c) + \frac{1}{t^2},$$

holds with probability $\geq 1 - \frac{\delta}{4K}$. By the union bound, it can be further proved that

$$\forall t \geq 1, \forall i \in [K], \quad |f_i(x_t^*, c) - \mu_{t-1}^i([x_t^*]_t, c)| \leq \beta_t^{\frac{1}{2}} \sigma_{t-1}^i([x_t^*]_t, c) + \frac{1}{t^2},$$

holds with probability $\geq 1 - \frac{\delta}{4}$. This further implies that

$$\inf_{Q \in \mathcal{U}(P_t)} \mathbb{E}_{c \sim Q}[\boldsymbol{s_t}^\top \boldsymbol{f}(x_t^*, c)] - \inf_{Q \in \mathcal{U}(P_t)} \mathbb{E}_{c \sim Q}[\boldsymbol{s_t}^\top (\boldsymbol{\mu_{t-1}}([x_t^*]_t, c) + \beta_t^{\frac{1}{2}} \boldsymbol{\sigma_{t-1}}([x_t^*]_t, c))]$$

$$\leq \inf_{Q \in \mathcal{U}(P_t)} \mathbb{E}_{c \sim Q}[\boldsymbol{s_t}^\top \boldsymbol{f}(x_t^*, c)]$$

$$- \inf_{Q \in \mathcal{U}(P_t)} \mathbb{E}_{c \sim Q}\left[\boldsymbol{s_t}^\top \left(\boldsymbol{\mu_{t-1}}([x_t^*]_t, c) + \beta_t^{\frac{1}{2}} \boldsymbol{\sigma_{t-1}}([x_t^*]_t, c) + \frac{1}{t^2}\right)\right] + \frac{1}{t^2}$$

$$\leq \frac{1}{t^2} \quad (31)$$

holds with probability at least $1 - \frac{\delta}{4}$. We then bound the last item shown in equation 29, which can be expressed by

$$\inf_{Q \in \mathcal{U}(P_t)} \mathbb{E}_{c \sim Q}[\boldsymbol{s_t}^\top (\boldsymbol{\mu_{t-1}}(x_t, c) + \beta_t^{\frac{1}{2}} \boldsymbol{\sigma_{t-1}}(x_t, c))] - \inf_{Q \in \mathcal{U}(P_t)} \mathbb{E}_{c \sim Q}[\boldsymbol{s_t}^\top \boldsymbol{f}(x_t, c)]$$

$$\leq \mathbb{E}_{c \sim Q_t^*}\left[\boldsymbol{s_t}^\top \left(\boldsymbol{\mu_{t-1}}(x_t, c) + \beta_t^{\frac{1}{2}} \boldsymbol{\sigma_{t-1}}(x_t, c) - \boldsymbol{f}(x_t, c)\right)\right], \quad (32)$$

where $Q_t^* \in \arg\inf_{Q \in \mathcal{U}(P_t)} \mathbb{E}_{c \sim Q}[\boldsymbol{s_t}^\top \boldsymbol{f}(x_t, c)]$. Similar to Lemma 3, with our selected $\beta_t$, it can be proved that

$$\forall c \in \mathcal{C}, \forall t \geq 1, \forall i \in [K], |f_i(x_t, c) - \mu_{t-1}^i(x_t, c)| \leq \beta_t^{\frac{1}{2}} \sigma_{t-1}^i(x_t, c) \quad (33)$$

holds with probability at least $1 - \frac{\delta}{4}$. With equation 33 holds, it can be further proved that

$$\inf_{Q \in \mathcal{U}(P_t)} \mathbb{E}_{c \sim Q}[\boldsymbol{s_t}^\top (\boldsymbol{\mu_{t-1}}(x_t, c) + \beta_t^{\frac{1}{2}} \boldsymbol{\sigma_{t-1}}(x_t, c))] - \inf_{Q \in \mathcal{U}(P_t)} \mathbb{E}_{c \sim Q}[\boldsymbol{s_t}^\top \boldsymbol{f}(x_t, c)]$$

$$\leq \mathbb{E}_{c \sim Q_t^*} \left[ \boldsymbol{s_t}^\top \left( \boldsymbol{\mu_{t-1}}(x_t, c) + \beta_t^{\frac{1}{2}} \boldsymbol{\sigma_{t-1}}(x_t, c) - \boldsymbol{f}(x_t, c) \right) \right]$$

$$\leq 2 \mathbb{E}_{c \sim Q_t^*}[\beta_t^{\frac{1}{2}} \boldsymbol{s_t}^\top \boldsymbol{\sigma_{t-1}}(x_t, c)].$$

Thus, based on the union bound, combining equation 29, equation 30, equation 31 and equation 32, we have that

$$r_t \leq 2 \mathbb{E}_{c \sim Q_t^*}[\beta_t^{\frac{1}{2}} \boldsymbol{s_t}^\top \boldsymbol{\sigma_{t-1}}(x_t, c)] + \frac{1}{t^2} \tag{34}$$

hold with probability at least $1 - \frac{\delta}{2}$. Moreover, we can find a constant $M$ such that

$$\mathbb{E}_{c \sim Q_t^*}[\beta_t^{\frac{1}{2}} \sigma_{t-1}^i(x_t, c)] \leq M \mathbb{E}_{c \sim P_t}[\beta_t^{\frac{1}{2}} \sigma_{t-1}^i(x_t, c)]. \tag{35}$$

Applying Lemma 5, it can be proved that for any $i \in [K]$,

$$\sum_{t=1}^T \mathbb{E}_{c \sim P_t}[s_t^i \beta_t^{\frac{1}{2}} \sigma_{t-1}^i(x_t, c)] \leq 2 \sum_{t=1}^T s_t^i \beta_t^{\frac{1}{2}} \sigma_{t-1}^i(x_t, c_t) + 8 \log \left( \frac{24K}{\delta} \right) \tag{36}$$

holds with probability at least $1 - \frac{\delta}{4K}$, where $s_t^i$ is the i-th element of $\boldsymbol{s_t}$. By the union bound, it can be further shown that

$$\forall i \in [K], \sum_{t=1}^T \mathbb{E}_{c \sim P_t}[s_t^i \beta_t^{\frac{1}{2}} \sigma_{t-1}^i(x_t, c)] \leq 2 \sum_{t=1}^T s_t^i \beta_t^{\frac{1}{2}} \sigma_{t-1}^i(x_t, c_t) + 8 \log \left( \frac{24K}{\delta} \right) \tag{37}$$

holds with probability at least $1 - \frac{\delta}{4}$. According to the information bound in Lemma 2, the Cauchy–Schwarz inequalit and similar to equation 28, it can be shown that

$$\sum_{t=1}^T \boldsymbol{s_t} \boldsymbol{\sigma_{t-1}}(x_t, c) \leq \sqrt{\sum_{t=1}^T \|\boldsymbol{s_t}\|_2^2 \sum_{t=1}^T \sum_{i=1}^K \sigma_{t-1}^{i2}(x_t, c)} \leq \sqrt{\sum_{i=1}^K \frac{2\gamma_T^i T}{\log(1 + \sigma_f^{-2})}}. \tag{38}$$

Combining equation 34, equation 35, equation 36, equation 38, the fact that $\beta_t$ is increasing, and by the union bound, we can prove that

$$\sum_{t=1}^T r_t \leq 4M \sqrt{\sum_{i=1}^K \frac{2\beta_T \gamma_T^i T}{\log(1 + \sigma_f^{-2})}} + 16MK \log \left( \frac{24K}{\delta} \right) + \frac{1}{t^2}$$

holds with probability at least $1 - \delta$. This completes the proof.

# D NUMERICAL EXPERIMENTS

All the random seeds are set from $100 - 104$ (Huang et al., 2024). For the context, we uniformly discretize the space into the size of $\lceil |\tilde{\mathcal{C}}|^{1/D} \rceil^D$, where $D$ is the dimension of the context variable. We consider the TV and $\chi^2$-DRO problems with a fixed uncertainty radius $\rho = 0.1$. We use the CVXPY Python package (Diamond & Boyd, 2016) to find the optimal dual variable in the DRO problem. The update of the Gaussian models and the solutions of the acquisition function are obtained by Botorch (Balandat et al., 2020). We share the same model parameter as Huang et al. (2024), where $\sqrt{\beta_t}$ is set to 1.5. For the optimize acqf function in Botorch, the num restarts is set to 10 and raw samples is set to 128. All Gaussian processes are calculated in one Nvidia H100, and all CVXPY are computed in AMD EPYC Milan 7713 CPUs. Here are the details of the functions used in the experiments.

### D.1 SINGLE-OBJECTIVE OPTIMIZATION

**Ackley function:** the input of this function is a 2-dimensional variable $x \in [0, 1]^2$ and 1-dimensional context variable $c \in [0, 1]$. We uniformly discretize the space into the size of $|\tilde{\mathcal{C}}|$ and function is expressed by

$$f(x, c) = ae^{-b\sqrt{\frac{1}{3}\left(\sum_{i=1}^{2} \tilde{x}_i^2 + \tilde{c}^2\right)}} + e^{\frac{1}{3}\left(\sum_{i=1}^{2} \cos(h\tilde{x}_i) + \cos(h\tilde{c})\right)} - a - e,$$

where $a = 20, b = 0.2, h = 2\pi$ are some constants and $\tilde{x} = 65.536x - 32.768$, and $\tilde{c} = 65.536c - 32.768$. The reference distribution $P_s$ is set to the uniform distribution over the discrete set.

**Modified Branin Williams (2000):** the input of this function is a 2-dimensional variable $x \in [0, 1]^2$ and 2-dimensional context variable $c \in [0, 1]^2$. We uniformly discretize the space into the size of $\lceil |\tilde{\mathcal{C}}|^{1/2} \rceil^2$ and function is expressed by

$$f(x, c) = -\sqrt{h(15x_1 - 5, 15c_1) \cdot h(15c_2 - 5, 15x_2)},$$

where $h(u, v) = a(v - bu^2 + cu - r)^2 + s(1 - t)\cos(u) + s$ with $a = 1, b = \frac{5.1}{4\pi^2}, c = \frac{5}{\pi}, r = 6, s = 10$ and $= \frac{1}{8\pi}$. The reference distribution $P_s$ is set to the uniform distribution over the discrete set.

**Hartmann:** the input of this function is a 5-dimensional variable $x \in [0, 1]^5$ and 1-dimensional context variable $c \in [0, 1]$. We uniformly discretize the space into the size of $|\tilde{\mathcal{C}}|$ and function is expressed by

$$f(x, c) = -\sum_{i=1}^{4} \alpha_i \exp\left(-\sum_{j=1}^{5} A_{ij}(x_j - P_{ij})^2 - A_{i6}(c - P_{i6})^2\right),$$

where $\alpha = (1.0, 2.0, 3.0, 3.2)^\top, A = [10, 3, 17, 3.50, 1.7, 8; 0.05, 10, 17, 0.1, 8, 14; 3, 3.5, 1.7, 10, 17, 8; 17, 8, 0.05, 10, 0.1, 14]$ and $P = 10^{-4}[1312, 1696, 5569, 124, 8283, 5886; 2329, 4135, 8307, 3736, 1004, 9991; 2348, 1451, 3522, 2883, 3047, 6650; 4047, 8828, 8732, 5743, 1091, 381]$. The reference distribution $P_s$ is set to the uniform distribution over the discrete set.

**Real-World problem (Cakmak et al., 2020; Nguyen et al., 2021b; Huang et al., 2024):** the input of this function is a 3-dimensional variable $x \in [0, 1]^3$, which represents the risk and trade aversion parameters, and holding cost multiplier in the portfolio optimization system. The 2-dimensional context variable $c \in [0, 1]^2$ denotes the bid-ask spread and borrowing cost. The objective function is set to the posterior mean of a Gaussian process fitted by 3000 samples.

The results for TV and $\chi^2$-DRO problems are shown in Fig. 3 and 4. From the results, we can observe that our methods introduce sublinear regrets, and the size of the context set has little impact on the model performance.

### D.2 MULTI-OBJECTIVE OPTIMIZATION

We conduct experiments on Ackley, Modified Branin, and Hartmann functions in a multi-objective optimization setting. In each experiment, there are 2 functions to be optimized, and the scalarization parameter $s$ follows a uniform distribution. While the first objective is the same as in the single-objective setting, the shared context has a different impact on the second objective. For the Ackley function, we have that

$$f_2(x, c) = ae^{-b\sqrt{\frac{1}{3}\left(\sum_{i=1}^{2} \tilde{x}_i^2 + \tilde{c}^2\right)}} + e^{-\frac{1}{3}\left(\sum_{i=1}^{2} \cos(h\tilde{x}_i) + \cos(h\tilde{c})\right)} - a - e.$$

For the Modified Branin function, we have that

$$f_2(x, c) = -\sqrt{h(15x_1 - 5, 15(1 - c_1)) \cdot h(15(1 - c_2) - 5, 15x_2)}. \tag{39}$$

For the Hartmann function, we have

$$f_2(x, c) = -\sum_{i=1}^{4} \alpha_i \exp\left(-\sum_{j=1}^{5} A_{ij}(x_j - P_{ij})^2 - A_{i6}(1 - c - P_{i6})^2\right).$$

The results for TV and $\chi^2$-DRO problems are shown in Fig. 1, 2 and 5. From the results, we can observe that our methods introduce sublinear regrets, and the size of the context set has little impact on the model performance.

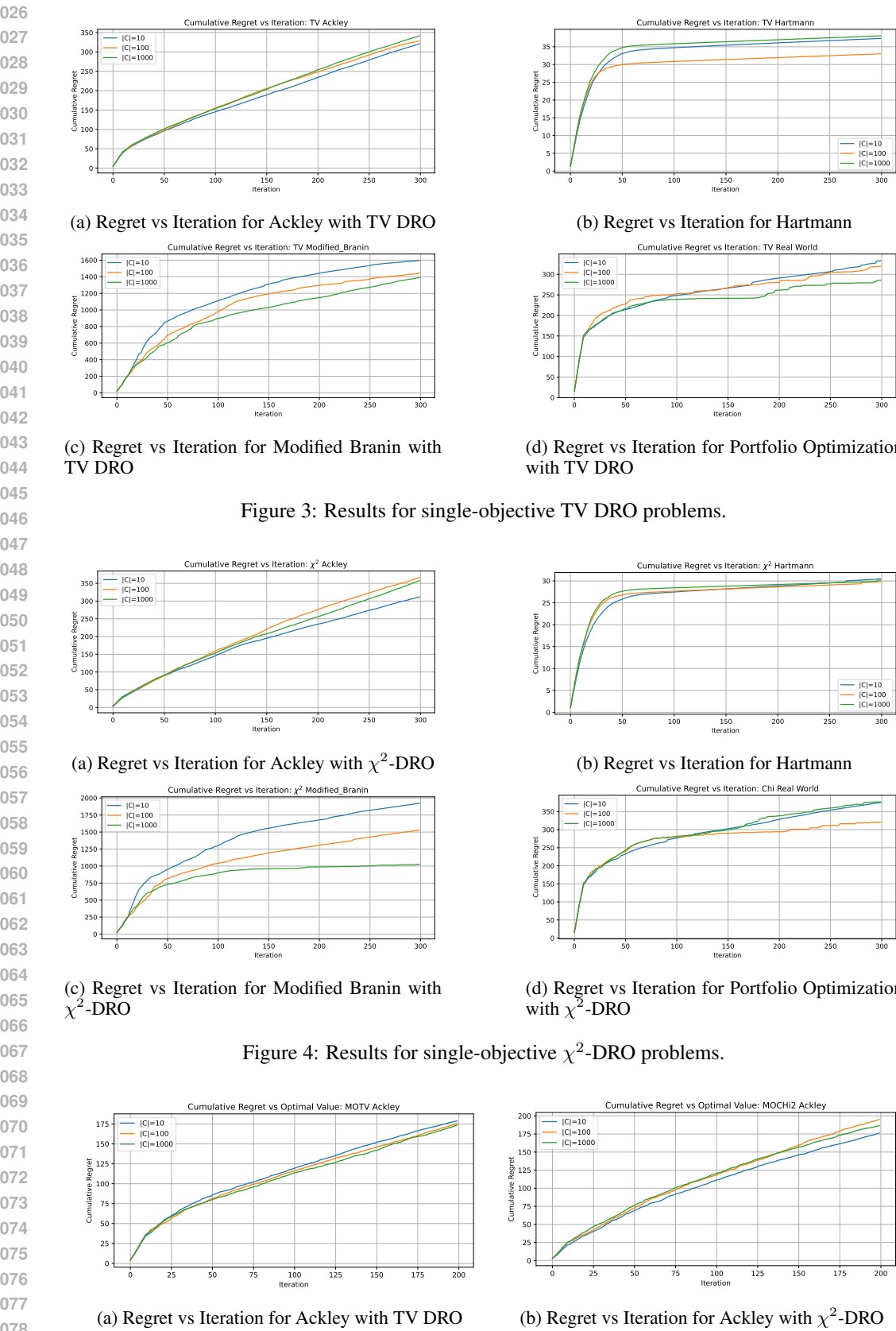

(a) Regret vs Iteration for Ackley with TV DRO

(b) Regret vs Iteration for Hartmann

(c) Regret vs Iteration for Modified Branin with TV DRO

(d) Regret vs Iteration for Portfolio Optimization with TV DRO

Figure 3: Results for single-objective TV DRO problems.

(a) Regret vs Iteration for Ackley with $\chi^2$-DRO

(b) Regret vs Iteration for Hartmann

(c) Regret vs Iteration for Modified Branin with $\chi^2$-DRO

(d) Regret vs Iteration for Portfolio Optimization with $\chi^2$-DRO

Figure 4: Results for single-objective $\chi^2$-DRO problems.

(a) Regret vs Iteration for Ackley with TV DRO

(b) Regret vs Iteration for Ackley with $\chi^2$-DRO

Figure 5: Results for multi-objective Ackley functions.

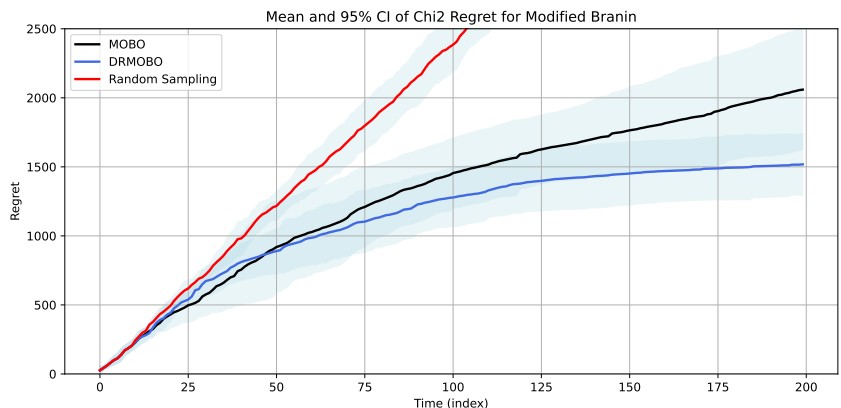

Figure 6: Regret vs Iteration for Modified Branin with $\chi^2$-DRO

## D.3 COMPARISON WITH BASELINES

In this section, we present the numerical results and compare our method with two baselines. The first baseline is random sampling, where solutions are uniformly sampled from the feasible set. The second baseline is the vanilla MOBO method, where the solution at each iteration t is selected

$$x_t \in \arg\sup_{x \in \mathcal{X}} \alpha'(x, \boldsymbol{s}_t) := \mathbb{E}_{c \sim P_t}\left[\boldsymbol{s}_t^\top \left(\boldsymbol{\mu_{t-1}}(x, c) + \beta_t^{\frac{1}{2}} \boldsymbol{\sigma_{t-1}}(x, c)\right)\right].$$

We conduct experiments on the synthetic functions such as Modified Branin, and Hartmann functions and one real-word problem, the newsvendor problem (Huang et al., 2024; Eckman et al., 2021), in a multi-objective optimization setting. In each experiment, there are 2 functions to be optimized, and the scalarization parameter $\boldsymbol{s}$ follows a uniform distribution. While the first objective is the same as in the single-objective setting, the shared context has a different impact on the second objective. The detailed function setup for the synthetic functions follows Appendix D.2. For the Newsvendor problem, we have a vendor that buys $x$ units liquid and sells to the customers with a demand of $c$. The cost for each unit is $s_0$, the sell price for each customer is $s_1$ and the unsold liquid can be returned for a price of $w$ for each unit. Thus the profit can be written as a function of decision $x$ and context $c$: $f(x, c) = s_1 \min\{x, c\} + w \max\{0, x - c\} - s_0 x$. We follow the same setting in Huang et al. (2024); Eckman et al. (2021), where $s_0 = 5, s_1 = 9$ and $w = 1$. The context $c$ follows a Burr Type XII distribution, where the PDF can be written as $p(c; \alpha; \beta) = \alpha\beta\frac{c^{\alpha-1}}{(1+c^\alpha)^{\beta+1}}$ with $\alpha = 2$ and $\beta = 20$. In the multi-objective setting, we consider an environment where the context $c$ changes, defining $f_1 = f(x, (c - 0.1)_+)$ and $f_2 = f(x, (c + 0.1)_+)$, where $c_+ = \max\{0, c\}$.

Figures 6–10 show the results compared with the proposed baselines. The uncertainty radius is set to $\rho = 0.5$ for synthetic functions and $1.0$ for real-world applications. All the context space is discretized into the size of $100$. Each experiment is repeated with random seeds from $100$ to $104$ to compute the mean and $95\%$ confidence intervals. As shown, the regret of random sampling increases linearly, indicating that this baseline fails to approach the optimal solutions. In all settings, the regret of our method grows more slowly over time compared with the vanilla MOBO, demonstrating that the proposed approach achieves higher efficiency and faster convergence.

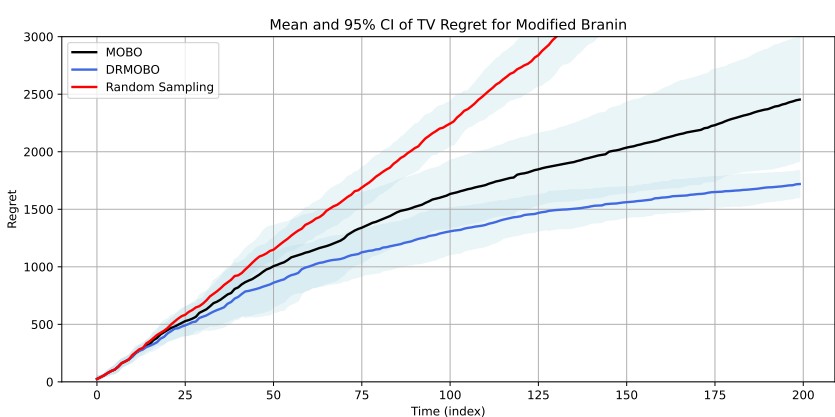

Figure 7: Regret vs Iteration for Modified Branin with TV-DRO

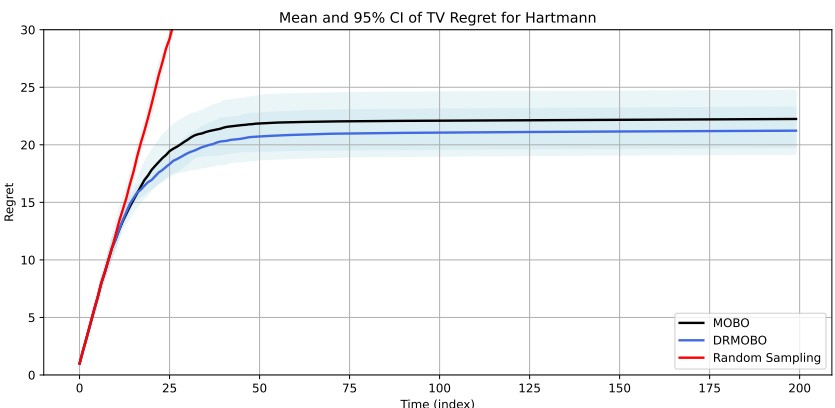

Figure 8: Regret vs Iteration for Hartmann with TV-DRO

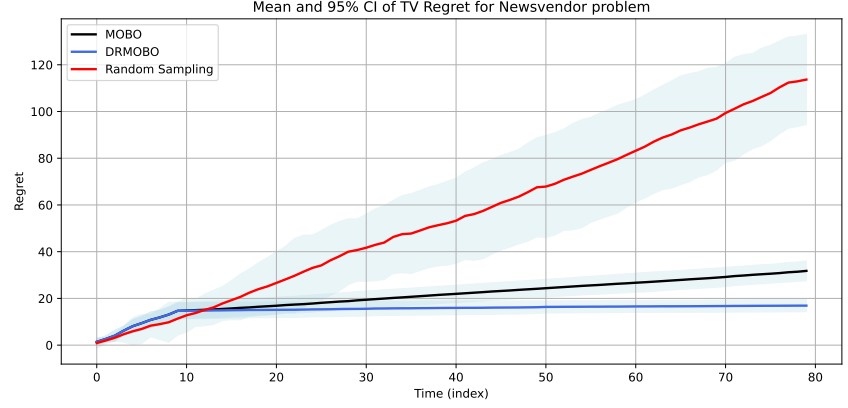

Figure 9: Regret vs Iteration for Newsvendor with TV-DRO

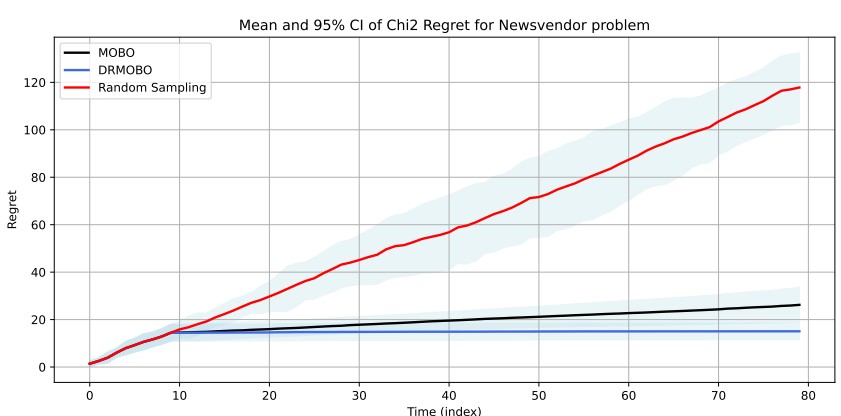

Figure 10: Regret vs Iteration for Newsvendor with $\chi^2$-DRO

