# OpenReview forum: "Distributionally Robust Bayesian Optimization: From Single to Multiple Objectives"
_ICLR.cc/2026/Conference — Submitted to ICLR 2026_

### Official Review · Reviewer_ffWs · 2025-10-23

**Soundness:** 3
**Presentation:** 3
**Contribution:** 3
**Rating:** 4
**Confidence:** 3

**Summary:**

The paper proposes a distributionally robust multi-objective Bayesian optimization (BO). The authors claim that most of studies distributionally robust BO is single objective except for only a few studies. Based on robust efficiency, the authors develop a scalarization and upper confidence based acquisition function, for which a dual-problem based formulation is derived. Further, the regret bound is derived based on weaker assumptions than existing studies.

**Strengths:**

The topic (distributionally robust MOBO) has not been widely studied, but I think it is potentially quite important topic. Overall, the paper is well-written and well-organized. Although I'm not familiar with this specific topic, the paper provides a comprehensible introduction of the background, motivation, and existing studies, by which the novelty of the paper is quite clear (even in ICLR submissions, there are surprisingly few papers that properly achieve this).

The theoretical analysis is seemingly reasonable and convincing (though I couldn't fully follow the entire proof).

**Weaknesses:**

A limitation by the linear scalarization is not clearly discussed.

The experiments are somewhat shallow and lack a baseline. I think that existing robust MOBO, even usual MOBO, and the random selection can be a baseline to demonstrate effectiveness of the proposed method. The goals of some of those methods  may be different from the proposed method, but for example, by comparing with usual MOBO methods, significance for considering a taylored approach for the target problem setting (distributionally robust MOBO) can be clarified. Without a baseline, efficiency is difficult to evaluate.

**Questions:**

The authors mention that the solution of linear scalarization is robust efficient. On the other hand, can any robust efficient solution be found by the optimizer of the linear scalarization (which is impossible in the case of the Pareto-frontier in usual multi-objective problems, i.e., so-called concave part of Pareto-frontier cannot be identified by linear scalarization)?

If the answer to the above question is no, can it be resolved by changing the scalarization function? Further, in that case, the regret analysis can also be extended to those other scalarizations?

Inatsu et al. 2024 also provide a theoretical analysis by using a distance based criterion between Pareto-frontiers, but the relation with the proposed analysis is not fully clear to me. In Table 1, the authors compare existing studies based on 'sublinear regret', but as mentioned above, I am not fully sure if the current definition of the 'regret' can capture the true efficiency about the identification of all the efficient solutions. Based on this consideration, when looking at Table 1, I wonder whether it is fair to regard the regret evaluation, which might not be perfect (may overlook a concave front), as an advantage, while not mentioning theoretical analyses using (perhaps) more fundamental metrics (directly comparing distance among Pareto-frontiers instead of comparing scalarized quantities). However, I do not have a deep understanding of this matter. Could you elaborate on this perspective?

---

> ### Author Response · Authors · 2025-11-21
>
> **W1**: We thank the reviewer for pointing this out. We agree that the use of linear scalarization introduces a limitation, as it can not enumerate the robust efficient points. However, we adopt this approach because it offers several advantages in our setting:
>
> It allows a tractable dual formulation of the $\varphi$-divergence DRO problem (Eq.9), leading to efficient optimization and theoretical analysis.
>
> The solutions obtained through linear scalarization are provably robust efficient, as established by Ehrgott et al, 2014.
>
> We {have} explicitly discussed this limitation in the revised manuscript (see line 252).
>
> **W2**:  Thanks for the reminder. We have added two additional baselines: random sampling and vanilla MOBO, to our experiments. The revised version now includes updated results with confidence intervals for the Modified Branin and Hartmann functions, which are presented in the Appendix D.3. As shown, the regret of random sampling increases linearly, indicating that this baseline fails to approach the optimal solutions. In all settings, our proposed DRMOBO outperforms the vanilla MOBO. In particular, for the Modified Branin function, the regret of our method grows more slowly over time compared with the vanilla MOBO, demonstrating that the proposed approach achieves higher efficiency and faster convergence.
> Due to time constraints, results for the Ackley function and the real-world problem will be provided soon.
>
> **Q1**: We thank the reviewer for this insightful question. The reviewer is correct that, as in standard multi-objective optimization, linear scalarization cannot recover the full Pareto frontier with non-convex objective functions. The same limitation holds in our robust setting — the solutions obtained through linear scalarization  can not enumerate the robust efficient points.
>
> **Q2**: Thanks for the question.
> The reason we focus on linear scalarization is because the obtained solutions are proved to be robust efficient. However, there is no such guarantees for other scalarizations such as Chebyshev and weighted Chebyshev. Also, extending the regret analysis to such alternative scalarizations would require substantial new theoretical development. In particular, nonlinear scalarizations such as Chebyshev introduce nonsmoothness and make the inner DRO problem more complex, and the resulting solutions may not always be guaranteed to be robust efficient.
>  Extending the framework by changing the scalarization function to fully recover all robust efficient solutions is nontrivial and remains an interesting direction for future research.
>
> **Q3**: We thank the reviewer for this insightful question. The analysis in Inatsu2024 is based on a distance-based criterion between Pareto frontiers and assumes that the contexts for each objective are independent. Under this assumption,
>     the worst-case distribution for each objective function can be calculated separately and it reduces the worst-case objective $f=(\inf_Q \mathbb E_Qf_1(x,c),..., \inf_Q \mathbb E_Qf_K(x,c))$ to a vector, making it relatively straightforward to identify the Pareto frontier.  In contrast, our formulation addresses a more challenging setting with shared contextual uncertainty across objectives, where directly characterizing the Pareto frontier is difficult. Instead, we focus on identifying robust efficient solutions that remain optimal under distributional uncertainty.
>
> The regret-based analysis we adopt follows the standard practice in Bayesian optimization and provides a tractable measure of convergence efficiency. A sublinear regret bound ensures that the solutions produced by our algorithm asymptotically approach a set of robust efficient solutions, offering a practical and theoretically grounded measure of learning performance under uncertainty.

---

> > ### Author Response · Authors · 2025-11-25
> >
> > Hi Reviewer ffWs,
> >
> > We have added more numerical results in both synthetic and real-world applications with baselines and confidence intervals. In all settings, the regret of our method grows more slowly over time compared with baselines, demonstrating that the proposed approach achieves higher efficiency and faster convergence.
> >
> > We hope our responses have adequately addressed your concerns and look forward to your new assessment.

---

### Official Review · Reviewer_u1go · 2025-10-28

**Soundness:** 2
**Presentation:** 3
**Contribution:** 2
**Rating:** 2
**Confidence:** 3

**Summary:**

This paper considers a distributionally robust multi-objective optimization problem, where the objective functions are expensive to evaluate and the ambiguity set (a set of context distributions) is defined by $\varphi$ divergence. For this problem, this paper proposes a UCB-type acquisition function, develops its efficient computation approach, and derives regret upper bounds in the Bayesian setting in which multiple objective functions follow Gaussian processes. Finally, the authors provide the simple benchmark results.

**Strengths:**

This paper is clearly well-written and easy for me to follow.
This paper appears to employ solid definitions and techniques from the distributionally robust (white-box) multi-objective optimization, distributionally robust Bayesian optimization, and multi-objective Bayesian optimization literatures.
The theoretical results also appear rigorous.

**Weaknesses:**

On the other hand, I simultaneously feel that there are several severe limitations listed below:
- The experimental results are weak. Since there is no baseline method, we cannot verify the effectiveness of the proposed method. Even if there is no existing method tailored for this problem setup, I would strongly encourage comparing with some vanilla baselines. On the other hand, I conjecture that the method in [Inatsu et al, 2024, Daulton et al., 2022] can be compared in some setting of $\varphi$.

- Although this paper is based on the definition of regret from [Paria et al., 2020], its definition slightly changes. Although Paria et al. (2020) take the expectation with respect to $s_t$, this paper does not take it as in Eq. (11). The original definition guarantees the proximity of the recommended solutions and Pareto optimal solutions by showing the upper bound of the expectation where $s_t$ varies in some distribution (typically a uniform distribution). Therefore, this change of the definition affects the interpretation of the theoretical result. I think that the definition in Eq. (11) is not suitable for multi-objective optimization since it can approach zero even if $s_t$ is always the same vector.

- This paper claims that it relaxes the condition on the radius of the ambiguity set. However, instead of that, this paper requires the condition that $P\_t(c) > 0$ for all $c$ such that $Q(c)$ can be positive. Since this is the additional assumption from the existing studies, it should be more explicitly discussed.

- The final regret upper bound is $O(K)$. However, I conjecture that it can be tightened as $O(\sqrt{K})$ by applying Cauchy--Schwarts inequality to $\sum\_{t=1}^T s_t \sigma\_{t-1}(x\_t, c) \leq \sqrt{ \sum\_{t=1}^T \| s_t \|_2^2 \sum\_{t=1}^T \sum\_{i=1}^K  \sigma^{i2}\_{t-1}(x\_t, c)} \leq \sqrt{ T \sum\_{t=1}^T \sum\_{i=1}^K  \sigma^{i2}\_{t-1}(x\_t, c)} $.

- Although Table 1 shows that Inatsu et al. (2024) is N/A regarding the sublinear regret,  Inatsu et al. (2024) provided Theorems 4.1 and 4.2, which claim that the algorithm must achieve an $\epsilon$-accurate solution within finite iterations. This is essentially the same result as the sublinear regret upper bound.

Overall, I feel that the lack of experimental comparison is the most important weakness.

**Questions:**

Please answer the above comments.
In addition, does the theorem imply that the algorithm can enumerate the robust efficient input points?

---

> ### Author Response · Authors · 2025-11-21
>
> **W1**: Thanks for the reminder. We have added two additional baselines: random sampling and vanilla MOBO, to our experiments. The revised version now includes updated results with confidence intervals for the Modified Branin and Hartmann functions, which are presented in the Appendix D.3. As shown, the regret of random sampling increases linearly, indicating that this baseline fails to approach the optimal solutions. In all settings, our proposed DRMOBO outperforms the vanilla MOBO. In particular, for the Modified Branin function, the regret of our method grows more slowly over time compared with the vanilla MOBO, demonstrating that the proposed approach achieves higher efficiency and faster convergence.
> Due to time constraints, results for the Ackley function and the real-world problem will be provided soon.
>
> **W2**: We thank the reviewer for this insightful comment. Our definition of regret in Eq.(11) indeed differs slightly from that of {Paria et al, 2020} and is the multi-objective version of Husain et al., (2023). The main difference is that {Paria et al, 2020} define the expected regret $\mathbb{E}_{\mathbf{s}_t}[r_t(\mathbf{s}_t)]$ by taking an expectation over random scalarization vectors $\mathbf{s}_t \sim P_s$, whereas our formulation analyzes the instantaneous regret for each sampled $\mathbf{s}_t$. In our case, the expectation inside Eq.(11) corresponds to the DRO formulation, while Eq.(11) itself represents the sampled robust regret. We consider $\mathbf{s}_t$ as a random variable and derive the regret bound with high probability, whereas {Pariaet al, 2020} establish their result in expectation.
>
> **W3**: We thank the reviewer for this insightful observation. The condition P(c)>0 is introduced to ensure that the term $\sum_t \sigma(x_t, c)$ , which involves the unknown context variable
> c, can be bounded by a function of $\sum_t \sigma(x_t, c_t)$.
> In contrast, existing DRBO studies such as {Huang et al, 2024} and {Tay et al, 2022} require the radius of the ambiguity set to shrink over time in order to guarantee sublinear regret. Our contribution is to remove this requirement and instead allow a constant-radius uncertainty set, which enables robustness against persistent distribution shifts in the context variable.
>
> We have clarified this point in the revision (see line 73).
>
> **W4**:  Thank you. the reviewer is correct that a Cauchy–Schwarz step can tighten the dependence on $K$. We have prepared the revision based on that.
>
> **W5**: We thank the reviewer for the insightful comment. We {have} revised Table~1 to clarify these differences and to better reflect the scope and assumptions of Inatsu et al. (2024) relative to our work. But we want to note that
>     the setting in Inatsu et al. (2024) assumes that the contexts for each objective are independent. Under this assumption,
>     the worst-case distribution for each objective function can be calculated separately and it reduces the worst-case objective $f=(\inf_Q \mathbb E_Qf_1(x,c),..., \inf_Q \mathbb E_Qf_K(x,c))$ to a vector, making it relatively straightforward to identify the Pareto frontier. In contrast, our formulation handles the more challenging case of shared contextual uncertainty across objectives, where it is challenging to identify the  robust efficient solutions.
> Moreover, the convergence guarantees in Inatsu et al. (2024) depend on an unknown function $q$, which is hard to derive an explicit convergence rate.
>
> **W6**: see W1
>
> **Q1**: We thank the reviewer for this question. Theorem 1 provides a sublinear regret bound, which implies that the sequence of evaluated solutions asymptotically approaches a set of robust efficient points. However, the theorem does not imply explicit enumeration of all robust efficient inputs. In fact, even for the non-convex multi-objective optimization with non-robustness setting, linear scalarization can not enumerate the Pareto set.

---

> > ### Author Response · Authors · 2025-11-25
> >
> > Hi Reviewer u1go,
> >
> > We have added more numerical results in both synthetic and real-world applications with baselines and confidence intervals. In all settings, the regret of our method grows more slowly over time compared with baselines, demonstrating that the proposed approach achieves higher efficiency and faster convergence.
> >
> > We hope our responses have adequately addressed your concerns and look forward to your new assessment.

---

> > > ### Comment · Reviewer_u1go · 2025-11-26
> > >
> > > I appreciate the additional experiments and clarifications.
> > >
> > > ### Experiments
> > > I still believe that the experimental results are weak.
> > > Since this paper focuses on the distributionally robust setting, comparisons with distributionally robust (multi-objective) optimization methods should be included.
> > > In particular, for example, why is there no comparison against [Daulton et al., 2022]?
> > > Moreover, I conjecture that, after the linear scalarization, we can apply a distributionally robust single-objective optimization method.
> > >
> > > On the other hand, I think that the additional experiments, which could significantly change the statement on empirical validation, are not preferable for the conference review.
> > > Therefore, I do not ask for any further experiments.
> > >
> > > ### Definition of regret
> > >
> > > I still believe that the definition of regret seems strange.
> > > In particular, the best achievable value $s_t^\top f(x)$ depends on $s_t$, which is made by users, in contrast to the definition by the expectation from [Paria et al., 2020].
> > > This makes an interpretation difficult.
> > > The regret analysis by [Paria et al., 2020] implies that the regret is bounded from above for any $s_t$ in terms of expectation.
> > > On the other hand, in the regret analysis of this paper, the case of $s_t \approx s$ for all $t$ can occur (this case is not excluded in the probability $\delta$).
> > > In such a case, the algorithm finds only one point in the Pareto front, though the paper claims as follows:
> > > > Thus, the goal of DRO in the multi-objective setting is to identify a set of robust efficient solutions.
> > >
> > > If only one Pareto solution is sufficient, there is no need to solve multi-objective optimization, since it suffices to solve single-objective optimization.
> > > I think this point should be more justified, or the analysis of the expectation should be performed.

---

> > > > ### Author Response · Authors · 2025-11-27
> > > >
> > > > **Experiments:**
> > > >
> > > > Our work is, to the best of our knowledge, the first to study distributionally robust multi-objective Bayesian optimization under uncertainty in a shared context, rather than robustness to input noise. This distinction is essential: [Daulton et al., 2022] address input-noise robustness using a (multivariate) VaR formulation, which is not a DRO problem.
> > > > If one ignores this conceptual mismatch and adapt MARS in [Daulton et al., 2022]  to a CVaR objective (which is a DRO problem), then the  main difference is the choice of scalarization: our method uses linear scalarizations, whereas MARS relies on Chebyshev scalarizations. As we note in Line 466, after linear scalarization the main algorithmic differences between our approach and existing single-objective DRO method [Huang et al.] lie in the selection of the context distribution and the DRO radius, rather than in the Bayesian optimization machinery itself.
> > > >
> > > > **Definition of regret**:
> > > >
> > > > As mentioned in Line 311, the scalarization vector $s_t$ is drawn from a distribution $P_s$ for each time $t$, exactly as in [Paria et al., 2020]. For general choices of $P_s$, e.g., the uniform distribution, the probability that all sampled $s_t$ form $t=0$ to T
> > > > are identical is extremely small. Thus, in practice, the algorithm explores multiple preference directions and collects multiple points. If one were to choose a degenerate distribution $P_s$ that places all its mass on a single scalarization vector, then it is unavoidable that both our method and the method of [Paria et al., 2020] would return only a single Pareto-optimal solution. This behavior is inherent to scalarization-based approaches and not specific to our regret definition.

---

### Official Review · Reviewer_Hmrd · 2025-10-30

**Soundness:** 2
**Presentation:** 2
**Contribution:** 2
**Rating:** 2
**Confidence:** 3

**Summary:**

The paper proposes a Distributionally Robust Multi-Objective Bayesian Optimization framework that extends robust Bayesian optimization to settings with multiple objectives and shared contextual uncertainty. It formulates robustness via a general $\omega$-divergence-based distributionally robust optimization model, encompassing well-known divergences such as KL, TV, and $\varepsilon^2$. The authors develop a random-scalarization algorithm that produces robust efficient (Pareto-optimal) solutions and establish a sublinear regret bound for this setting, without requiring the uncertainty-set radius to shrink over time. Theoretical guarantees are complemented by numerical experiments on synthetic benchmark functions showing consistent sublinear regret.

**Strengths:**

**Strengths:**

1) Multi-objective optimization problems appear in many real-world applications, yet existing distributionally robust Bayesian optimization methods focus almost exclusively on single-objective settings. This paper addresses this important gap by proposing a principled framework for robust multi-objective optimization.

2) The positioning of the paper within the literature is clear and well justified. In particular, Table 1 provides a concise and informative comparison with related works, helping the reader to understand the novelty and scope of the proposed approach.

3) The theoretical development is complemented by numerical experiments that effectively visualize the main results and confirm the sublinear regret behavior predicted by the analysis.

4) The framework is formulated for a general class of $\omega$-divergence DRO models, making it broadly applicable to various notions of robustness (e.g., TV, KL, $\varepsilon^2$, and Cressie--Read divergences).

**Weaknesses:**

**Weaknesses:**

1) The assumption of finitely supported context variables is highly restrictive. It remains unclear how the proposed approach scales to continuous or high-dimensional context spaces, which are the realistic cases in contextual Bayesian optimization.

2) The first six pages mainly summarize existing background on Gaussian processes and distributionally robust optimization. The main contributions appear only from Section 4 onward, and it is not clear what is novel in Section 3, as most of the material seems to restate known results.

3) Assumption 2 is difficult to interpret and practically verify. The paper should clarify under what conditions this assumption holds and how it could be checked or enforced in applications.

4) Theorem 1 is hard to follow. Important quantities such as the set $\mathcal{X}_t$ and the constant $d$ are not properly introduced, which makes it challenging to understand the result and its implications.

5) Algorithm 1 requires solving the optimization problem (10) in each iteration. The authors claim that this can be done efficiently using CVXPY, but no justification or computational complexity analysis is provided to support this claim.

**Questions:**

**Questions:**

1) You write "equation 10 can be efficiently computed using the CVXPY Python package (Diamond \& Boyd, 2016)." What exactly does "efficiently computed" mean in this context, and why is the problem computationally tractable?

2) What is the set $P_s$ in Algorithm~1? Is it related to the reference distribution $P_t$ (e.g., $P_s = P_t$ for $t = s$), or is it an independent sampling distribution for the scalarization weights?

3) The reference for the duality formula of the $\phi$-divergence DRO (Equation~9) is incorrect. This result originates from:
\emph{A. Ben-Tal, D. Den Hertog, A. De Waegenaere, B. Melenberg, and G. Rennen, "Robust solutions of optimization problems affected by uncertain probabilities," Management Science, 59(2):341–357, 2013.}
Please cite the original source.

4) Before Definition~1, are you not simply introducing the Minkowski sum? Please clarify this connection explicitly.

---

> ### Author Response · Authors · 2025-11-21
>
> **W1**: This assumption is introduced solely to enable a tractable theoretical analysis. In practice, for continuous context spaces, the proposed method remains applicable by discretizing the context domain. To the best of our knowledge, our work presents the first MOBO framework for the general $\varphi$-divergence DRO problem with shared contexts, designed to obtain robust efficient solutions.
>
> In the theoretical analysis, bounding the term $\sum_t \sigma(x_t, c)$ is challenging when the context variable $c$ is continuous and unknown. By assuming finitely supported contexts, this term can be bounded by a function of $\sum_t \sigma(x_t, c_t)$, which allows us to apply the information bound in Lemma 2 and derive a rigorous regret bound in the multi-objective setting.
>
> **W2**: We believe that Section 3 is both important and necessary for clearly presenting our problem formulation. In fact, it shows the core technical contribution of our work, where we formally define the distributionally robust multi-objective optimization (DRMO) problem and develop the corresponding algorithmic framework.
>
> Specifically, Section 3 introduces the complete formulation of the DRMO objective under shared contextual uncertainty (Eqs. (7)–(9)) and provides the first dual representation for general $\varphi$-divergence DRO with shared context in the multi-objective setting.
>
> Furthermore, Section 3 lays the foundation for our algorithmic design: by integrating random scalarization with the derived dual formulation, we obtain the proposed DRMOBO algorithm (Algorithm 1). This algorithm enables efficient optimization by sampling scalarization vectors, solving the dual DRO subproblem, and iteratively updating the Gaussian-process surrogate. Together, these components unify several classical DRO variants (e.g., TV, KL, $\chi^2$, and Cressie–Read divergences) within a single, generalizable framework that has not appeared in prior works such as Huang et al. (2024) or Paria et al. (2019).
>
> **W3**: Assumption 2 is a standard smoothness condition in Gaussian Process (GP)–based Bayesian optimization (e.g., Srinivas et al., 2009; Paria et al., 2020).
>
> Intuitively, the assumption requires that the objective functions do not vary too abruptly with respect to the decision variables—that is, their partial derivatives are bounded with high probability. This guarantees that the GP posterior mean and variance evolve smoothly, allowing the theoretical regret bound to hold uniformly across the domain.
>
> In practice, this condition holds for stationary kernels $k(x, x') = k(x - x')$ which are four times differentiable, such as the Squared Exponential and Matérn kernels with $\nu > 2$ (see Section~5.2 of rinivas et al., 2009), while it is violated for the Ornstein--Uhlenbeck kernel (Matérn with $\nu = 1/2$; a stationary variant of the Wiener process).
>
> We have clarified this intuition and explicitly state in the revision (see Line 349).
>
> **W4**:  The constant $d$ is defined in {Line 149} as the dimensionality of the decision space $\mathcal{X}$. In Theorem 1, we only require the scalar quantity $|\mathcal{X}_t|$, which denotes the cardinality of $\mathcal{X}_t$,
>     the $\epsilon$-net constructed over the compact input domain $\mathcal{X} \subseteq \mathbb{R}^d$ at iteration $t$. We have moved the definition of $\mathcal{X}_t$ to line 363.
>
> **W5**: We would like to clarify that the optimization over $x$ in (10) is performed using the BoTorch framework, identical to standard Bayesian optimization. The only additional computation involves finding the optimal dual variables $\lambda$ and $\eta$ in (10) using CVXPY.
>
>    Unlike other DRO formulations such as those based on MMD or Wasserstein distances, problem (10) is a pure maximization rather than a max–min optimization, which substantially simplifies computation. Moreover, the dual objective is concave in $\lambda$ and $\eta$, and both variables are scalars. Consequently, the problem has a very small dimension and can be solved to optimality using modern convex solvers.
>
> **Q1**: See w5
>
> **Q2**: In Section 3 (line 310-314), we had the following discussions: "In the MOO setting, we begin by randomly sampling the coefficients $\mathbf{s}_t$ based on a known distribution $P_s$ with the support $\Lambda$ {(Paria et al., 2020)}. The solution of Equation (8) for each $\mathbf{s} \in \Lambda$ is guaranteed to be robust efficient, while the distribution $P_s$ serves only to assign probability density across the resulting solutions. In this paper, we take $P_s$ to be the uniform distribution."
>
> **Q3**: We have fixed the citation issue.
>
> **Q4**:  Yes, it is the Minkowski sum. we have clarified this in the revision.

---

> > ### Comment · Reviewer_Hmrd · 2025-11-21
> > **Response to Rebuttal**
> >
> > I would like to thank the authors for the detailed response to my points. I am sorry to bother you again. The following answers are still confusing to me:
> >
> > > This assumption is introduced solely to enable a tractable theoretical analysis.
> >
> > I do not understand this justification. First, what is a tractable theoretical analysis? Second, you cannot say in case this assumption does not hold we finitely discretize a continuous random variable. How would that work for, say, a Gaussian?
> >
> > > We believe that Section 3 is both important and necessary for clearly presenting our problem formulation. In fact, it shows the core technical contribution of our work, where we formally define the distributionally robust multi-objective optimization (DRMO) problem and develop the corresponding algorithmic framework.
> >
> > I see in Section 3 a duality formula, Equation 9, which is wrongly referenced and goes back to A. Ben-Tal, D. den Hertog, A. De Waegenaere, B. Melenberg, and G. Rennen, "Robust solutions of optimization problems affected by uncertain probabilities," Management Science, 59(2):341–357, 2013. Moreover, I see 4 examples. How can this be a core technical contribution?
> >
> > > Solving (10) "efficiently" with CVXPY
> >
> > Let me phrase my confusion here differently. CVXPY is a modeling framework / modeling language for convex optimization in Python; it does not say anything about how the problem is solved and why it is efficiently solvable. Please correct me if I am wrong.

---

> > > ### Author Response · Authors · 2025-11-21
> > >
> > > **Q1**:  By “tractable theoretical analysis,” we refer to derive a sublinear, closed-form regret bound for the  DRMOBO problem studied in this paper. By assuming finitely supported contexts, the challenging term $\sum_t \sigma(x_t, c)$ can be bounded by a function of $\sum_t \sigma(x_t, c_t)$, which allows us to apply the information bound in Lemma 2 and derive a rigorous regret bound in the multi-objective setting.
> > >
> > > **Q2**: We note that Algorithm 1 can be applied to continuous random variables through two approaches:
> > > - Sample-based approximation: For a continuous random variable $c$, Equation (10) in Algorithm 1 involves taking an expectation over the distribution $P$. Following Huang et al., (2024), we optimize the acquisition function in (10) using a sample-average approximation (SAA), where the expectation is approximated by Monte Carlo samples of $c.$
> > >
> > > - Discretization: Alternatively, a continuous distribution (e.g., a Gaussian) can be truncated to a finite interval and then discretized on a uniform grid. Increasing the number of support points improves the approximation accuracy to the underlying continuous distribution.
> > >
> > > From a theoretical perspective, for any $Q \in \mathcal{U}(P_t)$ and $c \in \mathcal{C}$, and  $Q\_t^* \in \arg\inf\_{Q \in \mathcal{U}(P\_t)} \mathbb{E}\_{c \sim Q}[f(x\_t,c)]$,
> > > it can be shown that $\frac{Q(c)}{P_t(c)} \le M$
> > > holds for discrete samples, leading to $\mathbb{E}\_{c \sim Q\_t^*}  [\beta\_t^{\\frac{1}{2}} \\sigma\_{t-1}(x\_t,c)]
> > > \leq M\mathbb{E}\_{c \sim P\_t} \beta\_t^{\frac{1}{2}} \sigma\_{t-1}(x\_t,c)$
> > >
> > >
> > > However, it is unknown that whether the above bound holds for continuous random variables, except in specific cases such as CVaR. While our framework can be directly extended to CVaR-based DRO in continuous settings, handling the term
> > > $
> > > \mathbb{E}\_{c\sim Q\_t^*}\left[\sqrt{\beta\_t}\\sigma\_{t-1}(x\_t,c)\right]
> > > $
> > > requires non-trivial additional analysis, which we leave for future work.
> > >
> > > **Q3**:  While this is a theoretical paper on the DRMOBO problem, Section 3 provides a key technical contribution by introducing our proposed algorithm and explaining its design motivations. Specifically, in Section 3 we extend the classical dual formulation to the multi-objective setting with shared contextual uncertainty and integrate it within a Bayesian optimization framework. The resulting formulation unifies multiple divergence families under a single framework (Algorithm 1) and constructs a new acquisition function in Algorithm 1 for multi-objective DRO (Eq. 10),  a formulation that, to the best of our knowledge, has not been explored in prior work.
> > >
> > > **Q4**: We thank the reviewer for this clarification. The reviewer is correct that CVXPY is only a modeling framework and does not itself guarantee computational efficiency. In our case, efficiency arises from the structure of the dual objective in Equation (10), which is a concave maximization problem in two scalar variables, in $\lambda$ and $\eta$. For such low-dimensional convex problems, standard optimization methods, such as the Ellipsoid method, can achieve an $\epsilon$-optimal solution with a step of $\mathcal O(n^2\log(\frac{1}{\epsilon}))$, where $n=2$ is the number of optimized variables. Therefore, the dual problem can be solved to optimality within negligible computational cost. We have revised the paper based on the discussion.

---

> > > > ### Comment · Reviewer_Hmrd · 2025-11-26
> > > > **Response to response**
> > > >
> > > > Thanks you for the clarifications. I am partially happy with it and would like to stick to my grade.

---

### Official Review · Reviewer_oiuE · 2025-10-31

**Soundness:** 3
**Presentation:** 3
**Contribution:** 2
**Rating:** 4
**Confidence:** 3

**Summary:**

This paper studies $\varphi$-divergence distributionally robust multi-objective Bayesian optimization. Unlike prior work (Inatsu et al., 2024) that assumes independent context variables across objectives, this work considers shared context variables when constructing the ambiguity set for robustness. Building on the dual formulation of φ-divergence DRO, the paper extends this to the multi-objective setting via random scalarization (Paria et al., 2020). The authors derive sublinear cumulative regret bounds and validate the algorithm on synthetic benchmarks under TV and $\mathcal{X}^2$-divergence settings, empirically demonstrates consistent sublinear growth of cumulative regret under varies discretization fineity.

**Strengths:**

- The shared-context DRMO setting is realistic and practically important.
- The proposed algorithm is in principle applicable to a broad class of $\varphi$-divergences (TV, KL, $\mathcal{X}^2$, CVaR), providing flexibility in modeling different robustness notions.
- The proposed algorithm comes with theoretical guarantee.
- Compared to Husain et al. 2023 and Tay et al. 2022, the analysis builds upon high probability bounds instead of the deterministic bounds on function values which is more sensible.

**Weaknesses:**

- As this paper mainly extended Huang et al., 2024 for the multi-objective setting in combination with Paria et al 2019, although spotting this gap is plausible, the approach is very straightforward.
- Limited experimental scope:
    - **Missing baselines**: While it is understandable that the problem setting is new in MOBO and there may not have any approaches that may be directly comparable, but some simple baselines (e.g., random sampling) could be considered.
    - **Repetition**: Why there is no repetition variance provided.
    - **Real-word problem**: I think there are also synthetic real-life problem (e.g., crashworthiness) could be considered to fullfill the experiment except for the current well-studied synthetic problem.

I would be happy to reconsider my score if the author could provide more persuading experimental results.

**Questions:**

- **Extension to other divergences**: It is also mentioned that some other divergence measure like MMD and WD can be considered as future work, since the dual form is actually agnostic of the shift measure, how difficult the extension of this approach to this setting?
- **Thompson sampling**: Since Thompson sampling is also considered in Paria et al 2019, is there any reason that is not considered to be extended as well in this setting?
- **Chebyshev Saclarization**: Is there a reason of not considering Chebyshev sclarization?

---

> ### Author Response · Authors · 2025-11-21
>
> **weakness 1**:  We appreciate the reviewer’s observation. While our work shares a similar motivation with Huang et al. (2024) in single-objective setting, our algorithm and theoretical analysis were developed independently; we only became aware of their paper during the preparation of our numerical experiments. For the single-objective case, the DRO formulation is indeed conceptually straightforward in both works; however, our theoretical analysis and methodological scope are substantially different.
> Specifically, we make three key nontrivial contributions:
> (1) We establish the first sublinear regret bound for multi-objective DRO without requiring a decaying uncertainty radius.
> (2) We propose a general $\varphi$-divergence framework beyond the TV-only setting in Huang et al. (2024).
> (3) To achieve this, we develop new analytical tools, since the information bound (Lemma 2)  cannot be directly applied in our setting.
>
> In contrast, Paria et al. (2019) addresses multi-objective Bayesian optimization (MOBO) without robustness. While several extensions to robust MOBO have been explored, our work addresses a more challenging and practically relevant scenario, where multiple objectives are influenced by a shared contextual uncertainty. To the best of our knowledge, this is the first MOBO framework for the general $\varphi$-divergence DRO problem with shared contexts, designed to obtain robust efficient solutions.
>
> **weakness 2**: Thanks for the reminder. We have added two additional baselines: random sampling and vanilla MOBO, to our experiments. The revised version now includes updated results with confidence intervals for the Modified Branin and Hartmann functions, which are presented in the Appendix D.3. As shown, the regret of random sampling increases linearly, indicating that this baseline fails to approach the optimal solutions. In all settings, our proposed DRMOBO outperforms the vanilla MOBO. In particular, for the Modified Branin function, the regret of our method grows more slowly over time compared with the vanilla MOBO, demonstrating that the proposed approach achieves higher efficiency and faster convergence.
> Due to time constraints, results for the Ackley function and the real-world problem will be provided soon.
>
> **Q1**: The extension to MMD and Wasserstein divergences has two main challenges. The first is computational tractability: both MMD- and WD-based DROs require solving an inner max-min problem over distributions in a reproducing kernel or transport space, which is considerably more complex than the maximization problem for $\varphi$-divergences (see Eq.~(9)). The second challenge lies in the theoretical analysis, as it would require additional work to bound terms such as $\sum_t \sigma(x_t, c)$ when the context variable $c$ is unknown or implicitly defined by the transport map. We plan to address these challenges in future work by leveraging recent advances in kernel-based and optimal-transport DRO formulations.
>
> **Q2**: We thank the reviewer for this thoughtful suggestion. In general, Thompson Sampling (TS) does not provide significant improvements over the Upper Confidence Bound (UCB) approach. For example, in the standard MOBO framework of Paria et al., (2019), TS and UCB methods achieve the same level regret performance.
>
>  In the standard MOBO (Paria et al., 2019), the random sample function $f'_t(x)$
> follows some Gaussian distribution and the decision is selected by $x_t=\arg \max_x f'_t(x).$
>
> However, in our robust setting, the objective is not $f(x)$ but involves an inner minimization over distributions $Q$. Thus the decision is selected by $x_t=\arg \max_x \inf_{Q\in \mathcal U(P)}[s^\top f'_t(x,c)]$.
>
> Solving this inner stochastic optimization problem $\inf_{Q\in \mathcal U(P)}[\cdot]$ at each iteration requires more computation and additional analysis. We therefore focus on the UCB-based approach, which provides tractable regret guarantees in this robust multi-objective setting.
>
> **Q3**: We use linear scalarization since Ehrgott et al. (2014) proved that all solutions it obtains are robust efficient, while no such theoretical guarantee exists for the Chebyshev scalarization.

---

> > ### Author Response · Authors · 2025-11-25
> >
> > Hi Reviewer oiuE,
> >
> > We have added more numerical results in both synthetic and real-world applications with baselines and confidence intervals. In all settings, the regret of our method grows more slowly over time compared with baselines, demonstrating that the proposed approach achieves higher efficiency and faster convergence.
> >
> > We hope our responses have adequately addressed your concerns and look forward to your new assessment.

---

### Meta-Review · Area_Chair_omdV · 2026-01-05

**Summary:**

Reviewers have concerns on novelty of algorithm design, assumptions, organization of writing, theoretical results. The author rebuttal addressed some of these concerns. For example, Assumption 2 is indeed an assumption used in Srinivas et al., ICML-2010, but I suggest the authors to fully cite this assumption in the revision. However, most reviewers also share big concerns on experimental results. For example, missing basic baselines and errorbars. The amount of revision work to address the experiment concerns goes against the accept threshold, and this work can be improved a lot from reorganization and polishing before the publication, therefore, I recommend Reject.

**Reviewer Concerns:**

Reviewers have concerns on novelty of algorithm design, assumptions, organization of writing, theoretical results. Most reviewers also share big concerns on experimental results. After reading the author rebuttal and the paper revision, experiment remain a big concern of this work.

**Reviewer Scores:**

Reviewer Hmrd and Reviewer u1go didn't change their scores after reading the author rebuttal. Unfortunately, Reviewer oiuE and Reviewer ffWs didn't reply to the author rebuttal. After reading the author rebuttal and the reviews unanimously leaning towards reject, I think they wouldn't raise their scores.

---

### Decision · Program_Chairs · 2026-01-26

Reject